# Reservoir Assessment Tool Version 3.0: A Scalable and User-Friendly Software Platform to Mobilize the Global Water Management Community

Sanchit Minocha[1], Faisal Hossain[1], Pritam Das[1], Sarath Suresh[1], Shahzaib Khan[1], George Darkwah[1], Hyongki Lee[2], Stefano Galelli[3], Konstantinos Andreadis[4], Perry Oddo[5]

[1]Department of Civil and Environmental Engineering, University of Washington, Seattle, WA 98105, United States
[2] Department of Civil and Environmental Engineering, University of Houston, Houston, TX 77204, United States
[3] Pillar of Engineering Systems and Design, Singapore University of Technology and Design, Singapore 487372
[4] Department of Civil and Environmental Engineering, University of Massachusetts, Amherst, MA 01003, United States
[5] NASA Goddard Space Flight Center, Science Systems and Application Inc, Greenbelt, MD 2077

*Corresponding Author*: Faisal Hossain (fhossain@uw.edu)

**Abstract.** In the modern world, dams and the artificial reservoirs behind them serve the increasing demand for water across diverse needs such as agriculture, energy production, and drinking water. As dams continue to proliferate, monitoring water availability influenced by reservoir operations is now of paramount importance. The Reservoir Assessment Tool (RAT) is a data-driven software that integrates satellite remote sensing with hydrological models, enabling the estimation of key reservoir parameters such as inflow, outflow, surface area, evaporation and storage changes. The earliest version of RAT (version 1.0) was set up for 1598 reservoirs around the world with limitations in functional robustness, updating frequency and scalability. Some of these limitations on updating frequency and functional robustness were addressed in version 2.0 that was later made operational for the inter-governmental agency of the Mekong River Commission. Recognizing the need for scalability to mobilize the global water management community to benefit from satellite remote sensing, we hereby introduce RAT version 3.0. This version is optimized for accelerating open collaboration among users for continuous improvement and customization of RAT to enable reservoir management breakthroughs. RAT 3.0 represents a wholesale overhaul from the previous versions to empower the global community of users and developers in the spirit of the open-source movement. RAT 3.0 allows reservoir monitoring advancements and new functional developments that can be freely exchanged and seamlessly integrated for continuous evolution of the software. A centralized web application has also been established to facilitate the storage and dissemination of global reservoir monitoring information along with comprehensive training resources. RAT 3.0 aspires to bridge the traditional practices of water management community with the capabilities of satellite remote sensing. The global impact of the software can be expected to increase as uptake spreads, enabling a more sustainable and equitable utilization of our planet's water resources.

**Key words:** Reservoirs, satellites, scalability, open-source, cloud computing, distributed modelling

## 1  Introduction

Over the past seven decades, the number of large dams constructed globally to manage water for various purposes, has increased 10-fold, with the current count exceeding 58,700 (ICOLD, 2020). Figure 1 provides a geographic representation of the major dams, revealing the extensive presence of dams around the world (Lehner et al., 2011). This increase in the number of dams is a direct result of the growing demand for water resources (Gleick, 2012), which is driven by population growth, urbanization, industrialization, and increasing irrigation (Kannan and Anandhi, 2020). Dams create artificial lakes called reservoirs for reliable water storage and an increasing need for irrigation and domestic and industrial water use (Baxter, 1977). From here on, the term 'reservoirs' will be used interchangeably with 'dams' and also to refer to these artificial lakes. Dams also play a major role in mitigating floods by attenuating river flow (Sordo-Ward et al., 2012) and in generating hydroelectricity, thus providing a cleaner energy alternative (Baxter, 1977; Biswas, 2012).

While dams have played a crucial role in providing water and energy to communities around the world, they also have a significant number of drawbacks that cannot be ignored. Dams alter downstream water conditions and sediment flow, change water temperatures, and obstruct the movement of organisms and nutrients (Poff and Hart, 2002). As dams disrupt the natural flow of rivers, they can exacerbate drought and flood conditions and negatively impact the aquatic habitat and ecosystem services. This can further affect the livelihoods of people, such as fishermen and farmers.

Given these potential downsides, monitoring dams' performance to ensure safe operation and mitigating negative effects is extremely important. Moreover, monitoring can also help optimize dam operations and maximize its benefits for all stakeholders. Traditional methods of monitoring, such as installing gauges, can be costly and time-consuming as it requires regular maintenance and data collection. In the past few years, satellite remote sensing has emerged as a powerful alternative for monitoring dams because of the wide range of advantages it serves (Bonnema and Hossain, 2017; Bonnema et al., 2021). It allows monitoring at a global scale and provides valuable data on the aerial extent and elevation of surface water in the reservoirs. These time-varying observations of the reservoir's water condition allow the derivation of storage change. Satellite remote sensing provides near real-time data, making it possible to respond to changes in reservoir conditions in a timely and efficient manner, allowing for more effective management. Remote sensing can also address the issue of limited availability of data on reservoirs and water usage in transboundary river basins, which is often exacerbated by political or logistical barriers (Hossain et al., 2007).

Most studies that analyse the reservoir's dynamic state of storage change and inflow/outflow fluxes using satellite remote sensing are limited to a specific locations or periods (Vu et al., 2021; Bonnema and Hossain, 2017; Zhang et al., 2014). There are also satellite-based reservoir monitoring tools that provide information about reservoir state like water level, but are often restricted in terms of spatial coverage, spatial and temporal resolution, or an exclusive community of expert users. For example, the Global Reservoirs and Lakes Monitor (G-REALM) database, developed by the U.S. Department of Agriculture's Foreign Agricultural Service (USDA-FAS), routinely monitors lake and

reservoir height variations every 10 days based on Jason series nadir altimeter data. The Hydroweb database, developed at the LEGOS laboratory (Laboratoire d'Etudes en Géophysique et Océanographie Spatiales) in Toulouse, France, contains time series of water levels over large rivers, lakes and wetlands at the global scale (http://hydroweb.theia-land.fr/). Most reservoir monitoring studies or systems that are available today assume simplified operating rules based on reservoir storage capacity rather than the time-varying nature of the reservoir's actual operations driven by changing hydrology (supply) and user needs (demand). In 2021, Biswas et al. (2021) addressed these gaps by introducing an assumption-free and satellite data-driven reservoir monitoring tool called Reservoir Assessment Tool (RAT) version 1.0 (RAT 1.0). This version was later upgraded with a few enhancements by Das et al. (2022) as RAT version 2.0 (RAT 2.0).

RAT helps water managers to track a reservoir's dynamic state comprising inflow, outflow, evaporation, storage change, storage level and extent. RAT 1.0 was operational for 1598 reservoirs worldwide, at a monthly update frequency that was used to generate monthly estimates of reservoirs' dynamic states (Biswas et al., 2021). Readers can still access the outputs of this original version at http://depts.washington.edu/saswe/rat_beta. In RAT 2.0, the updating frequency was improved to sub-weekly by leveraging data from multiple satellites. RAT 2.0 further enhanced the accuracy of storage change and outflow as Das et al. (2022) devised a more advanced multi-sensor surface water classification technique called 'TMS-OS.' RAT 2.0 operations have so far been primarily limited to the Mekong River basin and inter-governmental agency stakeholders, with scalability and reproducibility for the broader water management community remaining unexplored (Das et al., 2022). For an interested user with limited literacy on satellite remote sensing data, operationalizing RAT 1.0 or 2.0 for tracking reservoirs in a river basin of their choice requires significant training.

This paper describes an improved version of RAT (version 3.0), hereafter termed as RAT 3.0, driven by the need to lower the barrier of entry for first-time users and to improve scalability, robustness and reproducibility. RAT 3.0 is designed to be easy-to-apply and easy-to-operate within an open-source software architecture. Software that is built on these design features can potentially mobilize the global water community to explore, use and improve RAT in the spirit of open-source development similar to Linux development. RAT 3.0 can benefit not only governments, hydrologists, researchers, and stakeholders but also the individual user who relies on water resources for their livelihood, such as farmers, city residents, and fishermen. Access to near real-time information on the reservoir's dynamic state can empower the individual user or a community by first quantifying any potential negative impact of floods and droughts, preparing for them, and optimally utilizing available water resources. The high frequency operational feature of RAT 3.0 can help farmers make informed decisions on irrigation and crop planning, assist city residents in water conservation efforts and future usage planning, and aid fishermen in identifying the best fishing locations while avoiding areas with low water levels. Additionally, a scalable software like RAT 3.0 can be used to plan and optimize hydroelectricity generation (Chowdhury et al., 2022). These are just some of the many examples of what may be possible when a reservoir monitoring tool is made scalable and easy to operate. Our motivation for

developing the RAT 3.0 software is to accelerate breakthroughs in the broader field of managed water resources driven by the grassroots needs of local users worldwide.

The architecture of RAT 3.0 has been designed in such a way that it requires minimal user input with additional flexibility added for the more advanced users. It is more robust and less susceptible to data gaps or instability that satellite remote sensing systems can sometimes experience. Operational efficiency has also been improved as RAT 3.0 can remember the equilibrium state of the basin at each time step, making the need for model spin up unnecessary in a cron job. The directory structure for major and intermediate outputs of RAT 3.0 has been reorganized for better database management. With this release of RAT 3.0, the conventional water management community can be more empowered with the vantage of space by providing an easy-to-use, scalable software architecture to monitor reservoirs. We believe such a software has the potential to build a self-sustaining community of users who can actively contribute to the continuous improvement and upkeep of the reservoir monitoring software.

**2 Foundation of Reservoir Assessment Tool (RAT) 3.0**

**2.1 History and Purpose**

RAT is a framework that employs satellite-based remote-sensing data and hydrological modeling to estimate the outflow of a reservoir via a mass balance equation. To estimate the inflow, RAT uses meteorological data and hydrological modeling. In RAT 1.0 (Biswas et al., 2021), the hydrological model used was the Variable Infiltration

Capacity (VIC) version 4.2.d, which was later replaced with VIC-5 (Hamman et al., 2018) in RAT 2.0 by Das et al. (2022). To compute storage change in a reservoir, two inputs are required: the area elevation curve (AEC) and the change in the surface area or elevation of the reservoir. The AEC is also known as Area-elevation volume (AEV) or hypsometric curves. The AEC curve summarizes quantitatively the bathymetry of the reservoir and helps compute the inundated area of the reservoir for a given storage level or vice versa. Consequently, with two successive satellite

observations of water elevation or surface area, the storage change can be derived. In RAT 2.0, the user had to provide the AEC for all reservoirs, which could be retrieved from either in situ pre-dam topographic maps or an available digital elevation model (DEM). The change in the surface area of the reservoir is estimated by observing the alteration in the number of water-classified pixels of the satellite imagery within the reservoir's region of interest. Alternatively, if the reservoir's water surface elevation information is available, such as from nadir-looking radar altimeters, surface

area can be estimated from the AEC. The Penman Equation (Penman, 1948) is used in RAT 2.0 to calculate evaporation from a reservoir, leading to the estimation of reservoir outflow using the mass balance equation (Figure 2). Readers are encouraged to refer to Biswas et al. (2021) and Das et al. (2022) for a process-based understanding of RAT. This paper will hereafter focus mostly on the software model development aspect of RAT 3.0.

RAT was developed as a first-of-its-kind reservoir monitoring tool that could remotely monitor the changing dynamics of reservoirs, providing the global community with a comprehensive understanding of how they impact natural river flow. One of its original purposes was to serve as a decision-making tool for forming and evaluating water-sharing

agreements between countries in a river basin. Despite the success of RAT 1.0, the monthly frequency of updating for reservoir state, limited its use. Monthly estimates of reservoir outflow and storage fail to capture the operating patterns of a reservoir at finer scales, such as hydropeaking, often required for many decision-making needs of the water manager. RAT 2.0 was therefore developed based on feedback from stakeholders, particularly from member countries of the Mekong River Commission (MRC), to improve the updating frequency (to weekly) and improve the accuracy of outflow data. This resulted in better decision-making for stakeholder agencies. As a result, RAT 2.0 was the first major operational uptake by a large inter-governmental river agency such as the MRC (Das et al, 2022).

## 2.2 Why RAT 3.0?

RAT 3.0 has been developed to expand on the improvements made in the RAT 2.0 framework and apply them on a global scale to achieve RAT's original design goals. The provision of sub-weekly estimates of reservoir outflow can enable the user to gain insight into the operational patterns of dams and their effects on the aquatic environment. RAT 3.0 has been designed to appeal to a broader range of users beyond just government water agencies and large stakeholder communities, using a user-friendly setup that can run locally on any Unix machine for any reservoir worldwide. Input requirements have been significantly reduced in RAT 3.0, making it easier for non-hydrology communities such as stakeholders representing farmers and fishing communities to operate the tool. Wherever possible, familiarity with data and parameters related to hydrologic modelling or remote sensing is not required for the RAT 3.0 user. The open-access, scalable and user-friendly features of RAT 3.0 is designed to nurture a self-sustaining community of users who can actively contribute to the open-source development of the software, leading to further advancements, modifications, and breakthroughs in new frontiers for managing Earth's precious water resources. Figure 3 summarizes the technological evolution of RAT from version 1.0 to version 3.0.

The scalability that is now made structurally inherent in the RAT 3.0 modeling platform can now improve scientific inquiries on reservoir-regulated river systems, their environmental implications and broader sustainability impact. For example, RAT 3.0 can now facilitate the modeling of human-induced alterations to the global water cycle and the evolving climate using digital twin experiments. Today digital twin experiments are the staple currency for scientific attribution studies. These experiments are crucial for scientific attribution studies, particularly in understanding the extent to which alterations in surface water are influenced by human activities versus natural factors, which were earlier difficult to carry out. Now that RAT 3.0 lowers the barrier of entry for global set up under a variety of assumptions and simulated scenarios, the scientific community can easily carry out such digital twin experiments. For example, today with RAT 3.0, one can have two versions of global climate model running, one that uses RAT to model the world's dams and one without modelling dams; or two versions of RAT, one assimilating latest satellite data and other not. There can also be other digital twin experiments that can be used for evaluating diverse elements such as satellite missions, Digital Elevation Models (DEMs), and hydrological models. As an illustrative example, consider the application of RAT to model reservoir parameters in two scenarios, one integrating data from the recently launched Surface Water and Ocean Topography (SWOT) satellite mission and the other excluding such data. RAT

can be used to investigate and quantify how much benefit do such new scientific satellite missions bring to improve our understanding of the state of the art.

Such digital twins allow convenient scientific breakthroughs in delineating attribution, meeting the essential requirements of the scientific community engaged in environmental stewardship. This clearly represents one of the key underlying scientific motivations behind development of RAT 3.0.

Another distinctive strength of RAT 3.0 lies in its capacity to generate historical and contemporary data for various reservoir parameters, including inflow, outflow, and evaporation. This capability enhances our understanding of the intricate relationships between these parameters and other environmental variables. By exploring global-scale correlations, such as greenhouse gas emissions versus storage change, downstream river temperature versus outflow, and fish count versus water elevation, RAT 3.0 enables comprehensive studies and testing of hypotheses. Additionally,

the copious data produced by RAT can be leveraged to forecast extreme events, such as floods, thereby contributing to proactive and effective risk management. Therefore, with the release of RAT 3.0, we anticipate catalyzing the scientific community's efforts to address the identified knowledge gaps, thereby advancing our understanding of reservoirs and their intricate dynamics.

**2.3 The Framework of RAT 3.0**

The RAT 3.0 framework comprises 14 key steps that need to be executed successfully to complete a RAT run and set it up for a river basin (Figure 4). While these steps are independent in their execution, some of them may depend on others for input. For example, outflow calculation requires the prior execution of inflow estimation through VIC and its streamflow routing. Hence, users must ensure that any omitted step has already run or that its output does not serve as input for another step. This independent execution of steps is advantageous as it eliminates the need to rerun

previously executed steps during debugging or software enhancement exercises. Additionally, users have the flexibility to choose specific steps to execute, allowing them to run just individual steps, such as the surface area time series calculation instead of a complete RAT run. The list of steps is shown in Table 1. The workflow diagram in Figure 4 summarizes the comprehensive process employed by RAT 3.0, executing the 14 key steps to generate essential reservoir data, including inflow, outflow, storage change, and evaporation.

In Step 1 of RAT 3.0, global meteorological data is downloaded, which includes precipitation, minimum and maximum temperature, and wind speed, for the user-provided duration. The IMERG Late product (Huffman et al., 2014) is used to retrieve daily precipitation, while temperature and wind speed data are sourced from NOAA NCEP/Climate Prediction Center (Kalnay et al., 1996). The data is then scaled, aligned, and clipped to the region of

interest. In Step 2, all the meteorological data is combined into a single NetCDF file in the format required by MetSim (Bennett et al., 2020) for input. The spatial resolution of the data remains fixed at 0.0625° for the entirety of the RAT run. Additionally, Step 3 generates region-specific parameter files for MetSim if not supplied by the user.

After the parameter and input files are prepared for MetSim, Step 4 initiates the MetSim run to produce forcings that are reformatted for input into VIC. If the user has provided parameter files that extend beyond the basin's spatial extent, Step 5 creates VIC's parameter files. Next, in Step 6, VIC generates gridded runoff from the meteorological forcings, which is converted to the format required by the routing model. RAT 3.0 employs the same routing model as RAT 2.0. To prepare for Step 8, parameter files for the routing model are created in Step 7, including the basin's flow direction file and station file, if necessary. Step 8 involves executing the routing model to generate daily, monthly, and yearly streamflow files for each reservoir.

Step 9 of RAT 3.0 involves the creation of a shapefile for the basin's reservoir polygons using a global reservoir shapefile, if available. In Step 10, the TMS-OS algorithm (Das et al., 2022), and Google Earth Engine (GEE) Python API (Gorelick et al., 2017) are used to produce a surface area time series for each reservoir by efficiently using cloud computing. RAT 3.0 currently utilizes TMS-OS, which relies on data from at least one optical sensor and one Synthetic Aperture Radar (SAR) sensor. This functionality allows RAT to monitor reservoirs since 2015, aligning with the availability of Sentinel-2 and Landsat-8 data.

Step 11 extracts the water level time series for any reservoir that lies on the Jason-3 altimeter track. In Step 12, the area-elevation curve for each reservoir is generated using a digital elevation model, unless provided by the user. Step 13 first calculates the daily inflow for each reservoir using the streamflow files created in Step 8 by the routing model. The storage change time series for each reservoir is then estimated using its surface area/water level time series and area-elevation curve, implementing the trapezoidal rule. The Penman equation of Penman (1948) is then used to estimate the water loss from a reservoir in terms of evaporation. Lastly, Step 13 implements the mass-balance approach to estimate outflow from the reservoirs. Finally, all outputs of the RAT 3.0 are prepared as separate time series CSV files for each reservoir in Step 14.

**3 Advancements in RAT 3.0**

RAT 3.0 introduces several improvements in terms of scalability, robustness, efficiency, ease of use while retaining the power for the user to customize, and improve specific processes in the spirit of open-source software development (Figure 5). These advancements are in line with the software design vision to increase the applicability of RAT to diverse fields allied to managed water such as food, energy, fisheries, bio-diversity, drought, recreation etc. Table 2 outlines the enhancements introduced in RAT 3.0 as compared to RAT 2.0. One of the most notable improvements in RAT 3.0 is its ability to offer users a global database that can serve as a default configuration. This global database contains input files for VIC global parameters, river basin shapefiles, flow direction and dam geolocation (from GranD) which consequently makes it easier for the first-time user to set up RAT 3.0 at a basin of their choice without requiring much customization.

Also, with the introduction of RAT 3.0, we adopt the semantic versioning system (Preston-Werner, 2023) to define the names of future releases. In this system, version numbers adhere to the format "MAJOR.MINOR.PATCH". A

MAJOR version is incremented when changes are made that are not backward compatible, a MINOR version is incremented when new features and functionality are added in a backward-compatible manner, and a PATCH version is incremented for backward-compatible bug fixes. This approach ensures clarity and consistency in communicating the nature of each release.

## 3.1 Scalability

RAT 3.0 operates at the river basin level and can be set up for any river basin worldwide. Users have the flexibility to either select a basin from the global database, allowing RAT 3.0 to automatically set up for the reservoirs within that basin. Alternately, users can provide specific basin geometry and reservoir information for RAT to work with. Moreover, for large-scale analysis, users can select multiple basins simultaneously by utilizing a single configuration file in RAT 3.0. The software also simplifies the creation of region-specific parameter files through automation. The streamlined and automated pipeline of RAT 3.0 allows for global-scale usage, eliminating the need for user dependencies during the process. Notably, RAT 3.0 utilizes a service account to access GEE Python API and converts reservoir geometries into earth-engine objects in real-time. This advancement eliminates the requirement of pre-loading reservoir shapefiles as assets into the user's earth-engine account. This characteristic, in conjunction with RAT 3.0's software architecture featuring modular components and independent stepwise execution, enhances scalability and user-friendliness.

## 3.2 Robustness

RAT 3.0 conducts thorough validation of user input, which is provided via a configuration file, ensuring crucial checks such as the chronological correctness of start and end dates. The software adopts significant improvements in error handling, implementing a two-level log file system. The first log file (level 1) corresponds to the user's RAT execution, providing concise information on the successful execution of each step for each basin, along with any associated errors. The second log file (level 2) corresponds to a basin for which RAT 3.0 has been executed and offers a comprehensive description of the computations and processes undertaken by RAT for that basin. The level-1 log file also provides the authentication status of the earth-engine service account provided by the user.

In managing missing meteorological data within the designated execution period, the resilience of RAT 3.0 should be noted. The version 3.0 software proactively handles potential issues of file corruption resulting from internet connectivity problems during satellite data file downloads. It triggers an automatic re-download process to ensure data integrity. Users also have the flexibility to provide a climatological dataset specific to the basin, empowering RAT 3.0 to identify unphysical precipitation values that are sometimes produced by satellite data algorithms and substitute them with more realistic alternatives.

To guarantee execution stability, RAT 3.0 integrates robust exception-handling mechanisms that prevent the failure of one step from obstructing the progress of subsequent steps. While the level-1 log file records any encountered errors, it may not provide in-depth insights into the underlying cause. Therefore, users are advised to seek support

through the RAT discussion forum at https://github.com/UW-SASWE/RAT/discussions, where the engaged user and developer community can assist in diagnosing and resolving specific errors. Additionally, RAT 3.0 verifies the availability of at least one satellite imagery for each reservoir within the desired execution period before initiating computations in GEE, effectively avoiding potential errors thrown by its Python API.

### 3.3 Efficiency

RAT 3.0 improves computational performance and memory usage. To optimize resource utilization, RAT 3.0 leverages the dask Python library (Rocklin, 2015), enabling parallelization of meteorological data downloads and routing model execution for multiple reservoirs. Additionally, it incorporates a smart mechanism to avoid redundant operations by skipping the downloading and processing of files as well as the creation of parameter files that already exist from previous RAT runs. The configuration file of RAT 3.0 has been expanded to include new sections that correspond to the various steps outlined in the 'Framework of RAT 3.0' subsection. This enhancement allows users to selectively choose a subset of these 14 steps, enabling them to reduce the execution time of RAT based on their specific output requirements or to resume the process from a previous checkpoint.

RAT 3.0 offers automatic handling of the spin-up time required by the MetSim and VIC model, while still allowing users to utilize VIC's 'hot start' feature. In the hot start feature, VIC hydrologic model bypasses the spin-up period and initializes the soil parameters using an existing parameter file, which, in the case of RAT 3.0, represents the soil state conditions at the end of the duration of the last RAT execution. This time-saving capability allows users to seamlessly resume the RAT execution from the point where it last ended, unlike RAT 2.0, which required starting from the beginning to ensure an adequate spin-up period for VIC. This hot start feature brings down the computational burden of any operational cron job involving RAT 3.0. It is worth noting that employing VIC's 'hot start' feature necessitates a spin-up time for the routing model, which varies based on the basin size and the duration required for water to travel from upstream to downstream locations within the basin. RAT 3.0 effectively accounts for this spin-up time to ensure precise and accurate results.

Efficiency in memory usage has been achieved through the elimination of the need to save parameter files for each RAT execution. Moreover, users are offered the convenience of automatically deleting intermediate data while retaining only the final outputs. By default, RAT 3.0 merges newly generated output data with the previously stored results based on the date. However, users have the flexibility to execute RAT 3.0 by replacing the previous outputs, enabling a fresh start when desired.

### 3.4 Ease of Use

RAT 3.0 improves user-friendliness, particularly in the realm of data management. A major improvement is the separation of the data directory from the source code directory. Users are no longer required to keep the data directory in the same location as the source code. This separation proves advantageous when users need to store the data directory on a different disk to overcome memory limitations. Furthermore, RAT 3.0 recognizes the extensive

generation of intermediate outputs and parameter files before obtaining the final outputs. To streamline file
organization and ease navigation, the data directory has undergone restructuring and re-organization. This allows users
to effortlessly locate files associated with specific intermediate processes within RAT 3.0, such as MetSim, VIC, and
GEE.

This version of RAT also introduces a convenient built-in feature for operationalizing RAT 3.0, allowing users to
specify the desired latency. While the typical latency of meteorological data ranges from 1-2 days, it is advised to set
a latency of 3 days when operationalizing RAT 3.0. This ensures that RAT 3.0 obtains all the necessary input data
from various satellite data servers and executes smoothly, without encountering any errors. By setting a latency of 3
days, RAT 3.0 will initiate execution from the end of the previous RAT 3.0 run and cover the duration up to three
days prior to the current day. This approach guarantees that RAT 3.0 has access to the most up-to-date and reliable
input data, providing users with comprehensive results.

The introduction of the global database simplifies the input requirements for utilizing and operationalizing RAT 3.0
for first-time users. The expanded functionality of executing RAT 3.0 for multiple basins or selectively choosing
specific tasks (steps) enhances the versatility of RAT. In RAT 3.0, users only need to provide a single configuration
file and a secrets file to customize their preferences and provide the necessary credentials. To enhance user visibility
and troubleshooting capabilities, RAT 3.0 incorporates two-level logging, enabling users to track the progress of each
RAT execution and promptly identify and address any encountered errors or warnings. These combined features
significantly reduce the barrier of usage for first-time users, making RAT more accessible to the broader water
community.

For additional support and for downloading the RAT 3.0 software, a comprehensive user guide is available on the
RAT Global web app, described in the following section. A dedicated discussion forum (see section 3.2) has been
established to foster collaboration among RAT users, facilitating the open exchange of experiences, challenges, and
specific use cases.

**3.5 Global Database**

While the inclusion of the global database may not be considered a technical advancement specific to RAT 3.0, its
presence has brought about a transformative shift in the development of RAT 3.0. Without the global database, the
significance of other enhancements in RAT 3.0 would have been diminished. Leveraging various global datasets
outlined in Table 3, RAT 3.0 now possesses default inputs that enable the execution of RAT 3.0 for any river basin
worldwide for the first-time user. This approach allows the user to focus more on improving the skill of RAT
iteratively once the initial version is quickly set up and made functional.

The GRDC river basin shapefile [row 1 of Table 3] is utilized to generate a grid of the river basin for calculating
inflow using VIC and routing models. In addition, the boundary polygon from this shapefile is also employed to select

only those reservoirs and dams from the GRanD dam database of Lehner et al. (2011) [rows 2 and 3 of Table 3] that lie within the river basin. RAT is then executed for only these dams and calculates inflow, surface area change, evaporation, and outflow. The routing model relies on the flow direction file [row 4 of Table 3] to calculate inflow based on gridded runoff data obtained from VIC.

The DEM elevation raster from the Shuttle Radar Topography Mission (SRTM) [row 5 of Table 3] is used by RAT to create a domain parameter file required for the execution of MetSim. VIC Soil and Domain Parameter raster files [rows 6 and 7 of Table 3] offer global-scale and high-resolution information, respectively, enabling the creation of input parameter files required by VIC. Please note that the VIC Global parameters are uncalibrated. Therefore, users are required to calibrate intricate yet challenging-to-measure parameters, including soil depth and the variable
infiltration capacity parameter. Achieving an accurate match between simulated and observed inflow requires careful calibration efforts. For a comprehensive understanding of the calibration process and additional details, we recommend referring to the Usage Notes provided by Schaperow et al. (2021).

       Lastly, the EGM2008 geoid model [row 8 of Table 3] is used to extract reservoir levels from altimetry data. It is
important to note that while these datasets are provided as default inputs for RAT execution, users have the flexibility to override these options according to their preferences and select dam locations that are not in the GranD database. Additionally, if no AEC (area elevation curve) file is provided for a reservoir, the SRTM 30 m digital elevation model available in GEE is used to automatically estimate the AEC for the reservoirs in question (Farr et al., 2007).

**4 RAT – Global Web App: The Confluence of Visualization, Training and Access to RAT 3.0 Software**

An interactive web application for RAT 3.0 has been developed and can now be accessed at www.satellitedams.net. This platform serves as an informational hub for RAT and provides users with an interactive map to visualize and analyse data from numerous dams worldwide (Figure 6). In addition, this global web app offers a comprehensive 'HELP' section where users can access three types of educational and training resources. These are: 1) accessing the source codes from the RAT 3.0 GitHub page; 2) detailed software documentation hosted on ratdocs.io and 3)
comprehensive tutorial detailing how to execute each of the 14 key steps for successful execution of RAT 3.0. Those interested in deepening their understanding of the tool can explore the 'PUBLICATIONS' section to access relevant literature and enhance their knowledge.

       The web application has been carefully developed to cater to a wide range of users, including stakeholders, government
officials, and the general public. The key idea was to make it easy for users to access and visualize the dynamic outputs on reservoir state that are routinely generated by University of Washington developers to develop a first-cut understanding. Currently, the tool offers data for reservoirs located in the Texas-Gulf, Mesopotamia, and Southeast Asian River basins, with plans to include information on 1600 major dams worldwide that were originally monitored in RAT 1.0 in the near future. Once the tool becomes operational for most of the world's major reservoirs, users will
have access to data (with a 3-day latency) on the dynamic state reservoirs (hindcast and nowcast). We firmly believe

that every individual user or entity should have equal access to such information of our managed water resources. By making this data publicly available on the global web app, the developers have contributed their part to mitigating the lack of access to water information in transboundary and ungauged river basins.

We have implemented an advanced visualizer feature (Figure 7) to expand the capabilities of the web application and meet the diverse requirements of different entities and stakeholders. With this feature, users can extract meaningful information without the hassle of downloading the data, allowing for a more streamlined and efficient analysis process.

The advanced visualizer component allows users to conduct in-depth analyses using the reservoir data by plotting
multiple axes on a single interactive graph, such as discharge, surface area, and storage change (Figure 8). This functionality extends to any two reservoirs globally, providing the flexibility to perform comparative analyses (see Figures 9-10). For instance, users can compare the outflow of an upstream dam with the inflow of a downstream dam, shedding light on the dynamics between the two. Another scenario may involve plotting the surface area of a reservoir alongside its inflow, allowing users to gain a comprehensive understanding of the relationship between these reservoir
state variables (Figures 6-10).

**5 Applications: Leveraging RAT 3.0 in Practical Scenarios**

The versatile capabilities of RAT 3.0 extend across a range of critical hydrological and environmental applications, offering a valuable modelling platform for researchers and practitioners alike. The below mentioned use cases showcase the efficacy of RAT 3.0 in addressing complex challenges in diverse geographical settings. From tracking
extreme flooding in the mountainous terrains of Kerala, India, to the development of algorithms like ResORR for estimating regulated inflow demonstrates its prowess in hydrological modeling and analysis. This section provides a glimpse into the manifold applications of RAT 3.0, emphasizing its significance in advancing our understanding of impact of reservoir dynamics in different regions worldwide.

1.  Studying flood tracking in mountainous regions characterized by high precipitation and steep terrain, with a specific focus on the 2018 floods in Kerala, India using RAT 3.0 (Suresh et al., 2023). In this study, RAT 3.0 was able to clearly track extreme flooding in a highly mountainous and high precipitating region with perennial cloud cover and hydropower dams that compete with flood control. RAT3.0 was able to demonstrate that using the framework the timing of the peak flooding, the onset of sudden flooding and the
time to recede can all be captured by using only satellite data and RAT 3.0 (Figures 11 and 12).

       2.  Use of RAT 3.0 to assess the impact of the Belo Monte Dam on the Xingu River in the Amazon (Hossain et al., 2023). In this study, RAT 3.0 was able to pick up the historical hydrologic response to precipitation of the region during the post-dam period (after the Belo Monte dam was constructed).
       3.  Use of RAT 3.0 to develop algorithm (called ResORR) to estimate regulated inflow using VIC's natural inflow (which assumes the absence of upstream dams) with study site as Tennessee River basin (Das et al., 2023). In this study RAT 3.0 combined with modeling of regulation of flow by upstream reservoir storage was found able to consistently predict well the downstream inflow to a series of reservoirs (Figures 13 and
14).

4. Valuable insights provided by RAT into the effects of dams in the transboundary river basin of Mesopotamia (Hossain et al., 2023). In this study, RAT was developed, set up and operationalized over the Tigris Euphrates River basins exclusively to demonstrate to stakeholders such as Iraqi Ministry of Water Resources what is possible for transboundary reservoir monitoring using satellite data alone. Readers can see the operational system running at https://depts.washington.edu/saswe/rat. The scientific potential of this RAT rendition is that it brought various communities together to discuss how the Mesopotamian rivers could be restored and how the downstream marshland could be revised using techniques such as RAT when there is no in-situ data or publicly available modeling tools.

5. Analyzing the role of reservoirs on downstream river temperature using RAT 3.0 for Columbia River basin (Darkwah et al. 2023). In this study, RAT generated reservoir storage was able to improve the remote sensing-based surface water temperature estimation in the 50 km downstream reach of the Columbia River dams. With RAT derived storage change, there was about 50% reduction in RMSE of temperature prediction (Figure 15).

## 6 Discussion and Conclusion

In a world where dams continue to be constructed or maintained at a rate faster than they are dismantled, it is important to acknowledge that dams are here to stay in the foreseeable future with their inherent advantages and disadvantages. Our focus here is on mitigating the negative impacts associated with these structures by levelling the playing field of access to information on the dynamic state of the world's reservoirs. Timely understanding of how dams and reservoirs affect natural river flow and other important water parameters is crucial in achieving this goal. Real-time monitoring of reservoirs is also essential for effectively addressing immediate threats such as floods and droughts. RAT 3.0 makes such monitoring possible with satellite remote sensing missions that have emerged as an indispensable vantage for studying reservoir operations and their impact on surrounding areas.

In this paper, we have described the redesigned software architecture of RAT 3.0 that can be executed on any Unix operating system and even on supercomputers by making use of parallelization. The operating system constraint arises from the VIC hydrological model's exclusive compatibility with Unix OS. Consequently, a standard laptop with 8 GB RAM, 4 cores, and a 512 GB hard disk is sufficient for running RAT 3.0, with the computational time contingent on the size of the river basin.

The modular design of RAT 3.0 with 14 self-contained steps makes it possible for a user to execute RAT only for a desired set of tasks. This enhances the user-friendliness of RAT 3.0 and lowers the barrier of entry for the first-time user. Several improvements, like multiple log files, reduction of input requirements, and provision of customization using a config file, have been done to maximize user-friendliness. Error handling has been improved and RAT 3.0 is now more robust when input data is unstable or missing. RAT 3.0 is computationally more efficient by adopting parallelization during downloading and processing meteorological data and the automatic deletion of intermediate files. Finally, RAT 3.0 has been developed with global scalability in mind so that it can be set up for any reservoir system worldwide. Scalability is the most important consideration for developing and nurturing a self-sustaining

community of users who can continuously improve such a software in the spirit of open-source development and open science.

RAT 3.0 simulation can span from a single day to multiple decades and is dictated by the hydrologic model's smallest time step and satellite observation frequency. The Tiered Multi-sensor- Optical/SAR (TMS-OS) algorithm, that

calculates reservoir storage change based Sentinel-1 SAR, Sentinel-2, and Landsat satellites, allows RAT 3.0 to operate from 2015 onwards. However, single-sensor functionality can be extended to the early 80s, utilizing Landsat 4 data as has been demonstrated by Biswas and Hossain (2021).

RAT 3.0 has already undergone extensive global testing across various basins by multiple users, as illustrated in Figure

16, covering a wide range of use cases. Notably, RAT 3.0 has been instrumental in studying flood tracking in mountainous regions characterized by high precipitation and steep terrain, with a specific focus on the 2018 floods in Kerala, India (Suresh et al., 2023). It has been utilized to assess the impact of the Belo Monte Dam on the Xingu River in the Amazon (Hossain et al., 2023) and to compute and compare regulated inflow with natural inflow in the absence of upstream dams in the Tennessee River basin (Das et al., 2023). RAT 3.0 has also provided valuable insights into

the effects of dams in the transboundary river basin of Mesopotamia (Hossain et al., 2023). Recently, RAT was upgraded by the Asian Disaster Preparedness Center (ADPC) from version 2.0 to 3.0 for the Mekong River basin with assistance from co-author Pritam Das (see https://ratmekong-servir.adpc.net/). This version currently serves as an operational system for the MRC stakeholders. Currently, RAT 3.0 is being employed to analyse the hydro-thermal history of the Columbia River in the Pacific Northwest and upgraded from version 1.0 for the Nile and Indus River

basins (Figure 16). The fact that RAT 3.0 is applied in various river basins, coupled with the reduced setup time to a week from 3-4 months (for RAT 2.0), highlights the scalability and user-friendly architecture of RAT 3.0.

RAT 3.0 does not represent the final word on satellite-based reservoir monitoring software development. There will undoubtedly be numerous unforeseen software hurdles or bugs that will emerge as the use of RAT 3.0 expands. New

satellite missions will emerge, and many current ones may become less relevant, requiring updating of certain components of the software architecture. One such satellite mission is the recently launched Surface Water and Ocean Topography (SWOT) mission that will produce more accurate reservoir storage level, area and change data. Based on simulation studies reported by Bonnema and Hossain (2019), we believe SWOT will "turbocharge" the accuracy and operational efficiency of RAT 3.0 (see also Hossain et al., 2023; 2022).


The impact of sedimentation on reservoir storage is a critical consideration addressed in RAT 3.0. The framework's modular design allows users to dynamically update the area-elevation relationship, ensuring accurate simulations even in the presence of sedimentation. Localized applications in the Mekong and Cumberland River basins have showcased the framework's adaptability in representing changes in bathymetry due to sedimentation (Das et al., 2022; Das et al.,

2023). However, extending this capability globally faces challenges due to the lack of time-varying global digital elevation models (DEM). For pre-2000 reservoirs, such as those covered by the Shuttle Radar Topography Mission

(SRTM), representing sedimentation requires additional parameterizations or satellite observations when the reservoir was at lower levels (Bonnema et al., 2017). Future research endeavors with RAT 3.0 will explore potential solutions, involving sediment trapping concepts and the integration of satellite geodetic observations, to enhance its global
scalability in addressing sedimentation challenges.

The calibration and choice of the hydrologic model play a crucial role in RAT's performance, as it relies on the accuracy of unregulated inflow capture, especially at the headwaters. A recent study (Das et al., 2023) highlights the importance of hydrologic model calibration up to the last upstream boundary point where flow remains naturalized.
Beyond this point, sensitivity to calibration decreases, emphasizing the significance of assimilating reservoir storage changes to account for regulated flow downstream using an appropriate regulation algorithm. Thus, RAT is purported to be hydrologic model agnostic. In its current architecture RAT 3.0 is not yet streamlined to allow easy swapping of the hydrologic model, select specific satellites and input data streams by the user. All these issues will no doubt be addressed in a future and minor upgrade of RAT 3.0. What is important to note is that with the wholesale overhaul of
the RAT software architecture in version 3.0, we now have the necessary design spirit of user-friendliness, scalability, open-access and open-sharing embraced to nurture a self-sustaining community of users who can continuously improve satellite-based reservoir monitoring based on grassroots needs of users.

The current applications of RAT 3.0 mark the advent of a new era, where the combined power of high-performance
computing, information technology and satellite earth observations enable us to deepen our understanding of dams and their ecological impacts. RAT 3.0 represents the vanguard of this progress, offered to the global community as the first stride towards advancing reservoir management practices and bringing them up to speed in the 21st century space age. As water demand continues to escalate, RAT 3.0 equips water resource managers and stakeholders with invaluable insights to make informed decisions and optimize the utilization of scarce water resources. With RAT 3.0,
it is our hope that the deep-seated disparities among nations pertaining to transboundary rivers can be drastically reduced, contributing to a more harmonious world. We believe the day is not far when the transformative power of the open source and collaborative software development spirit of RAT 3.0 will touch the lives of all, ensuring equal access to information on the reservoir's dynamic state.

**Code Availability:** The RAT 3.0 software code is available for download from the HELP menu of the RAT 3.0 global web app at http://www.satellitedams.net Alternately, the code can also be downloaded from https://zenodo.org/record/8268138

**Data Availability on RAT 3.0 Global Database:** https://zenodo.org/record/8267708

**RAT 3.0 Training Resources Availability:** Software documentation on RAT 3.0 can be accessed from
http://ratdocs.io User manual is available at: https://depts.washington.edu/saswe/rat/user_manual/RAT-3.0_User_Manual.pdf

**Author Contribution:** Sanchit Minocha: research design, analysis, testing and writing; Faisal Hossain: research design, writing and editing; Pritam Das: research design, analysis, editing. All other co-authors: writing and editing

**Competing Interest:** The authors declare that they have no conflict of interest

**Acknowledgements:** The work on the development of the RAT 3.0 software architecture was generously supported by the NASA Applied Science Program through grants 80NSSC22K0918 (Water Resources) and 80NSSC23K0184 (Capacity Building) awarded to the second and corresponding author. The first author was supported from NASA Applied Science Program grant 80NSSC22K0918, National Science Foundation Graduate Traineeship program on Future Rivers and the Ivanhoe Foundation Fellowship.

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

| Step Number | Step Name |
|---|---|
| 1 | Downloading and Pre-processing of meteorological data |
| 2 | Pre-processing of data and preparation of MetSim Input |
| 3 | Preparation of MetSim Parameter Files |
| 4 | Running MetSim & preparation of VIC input |
| 5 | Preparation of VIC Parameter Files |
| 6 | Running of VIC and preparation of Routing input |
| 7 | Preparation of Routing Parameter Files |
| 8 | Running Routing |
| 9 | Preparation of parameter files for Surface Area Calculation |
| 10 | TMS-OS Surface Area Calculation from GEE |
| 11 | Elevation extraction from Altimeter |
| 12 | Generating Area Elevation Curves for reservoirs |
| 13 | Calculation of Outflow, Evaporation, Storage change and Inflow |
| 14 | Conversion of output data to final format as time series |

**Table 1: List of the 14 steps along with their functions which can be executed by running RAT 3.0.**

| Aspect | | RAT 2.0 | RAT 3.0 |
|---|---|---|---|
| **SCALABLE** | a) | Limited to Mekong River basin and its dams | Generalized for any reservoir worldwide |
| | b) | High number of manual input requirements | Automatic input generation reduces number of inputs required from user |
| | c) | Manual work required to use assets in Google Earth Engine and create inputs | Does not use Google Earth Engine assets; automatic creation of inputs at the time of execution |
| **ROBUST** | a) | Does not handle missing meteorological data | Improved handling of missing data with automatic interpolation |
| | b) | Limited error handling; Difficult to debug with one log file | Enhanced error and exception handling; Easy to debug with 2 log files |
| | c) | Failure in one component disrupts entire execution | Modular architecture allows independent execution |
| **FRIENDLY USER-** | a) | Disorganized output files | Intuitive directory structure for organized outputs |
| | b) | All inputs were not provided through a configuration file | Single configuration file handles all inputs |
| | c) | Lack of operationalization feature | Built-in feature for operationalization that works with cron job |
| **EFFICIENCY** | a) | No hot-start feature, less efficient use of resources | Hot-start feature for efficient resumption of execution |
| | b) | Limited use of parallel processing within RAT Framework | Parallelization with Dask library for resource efficiency in RAT Framework |
| | c) | Accumulation of output files, inefficient memory use | User-selectable automatic deletion of intermediate output files |

**Table 2: Comparison of enhancements in RAT 3.0 as compared to RAT 2.0 in terms of scalability, robustness, user-friendliness and efficiency.**


| S. No. | Dataset Name | Brief Description | Additional Information | Reference |
|---|---|---|---|---|
| 1 | GRDC Major River Basins | Shapefile of all the major river basins' polygons in the world. | Version - 2nd Revised Edition, 2020; Src- The Global Runoff Data Centre | GRDC, 2020 |
| 2 | GRanD Dams | Shapefile of all the major dams' locations in the world. | Version - 1.3 ; Src - https://www.globaldamwatch.org | Lehner et al., 2011 |
| 3 | GRanD Reservoirs | Shapefile of all the reservoirs' polygons associated with dams in [2]. | Version - 1.3 ; Src - https://www.globaldamwatch.org | Lehner et al., 2011 |
| 4 | DRT Flow Direction | Global raster file providing the flow directions in WGS84 projection. | Resolution - 1/16º; Src- Numerical Terradynamic Simulation Group , University of Montana | Wu et al., 2011, 2012 |
| 5 | SRTM_30 Elevation | Global elevation raster file providing elevation in meters. | Resolution – 30 arc seconds ; Src - Scripps Institution Of Oceanography, University of California San Diego | Becker et al., 2009 |
| 6 | VIC Soil Parameters | Global VIC soil parameter dataset containing separate raster files for each continent in WGS84 projection. | Resolution - 1/16º; Spatial coverage from 60°S to 85°N; Version-1.6d ;Src- DOI 10.5281/zenodo.3475601 | Schaperow et al., 2021 |
| 7 | VIC Domain Parameters | Global VIC domain parameter dataset containing separate raster files for each continent in WGS84 projection. | Resolution - 1/16º; Spatial coverage from 60°S to 85°N; Version-1.6d ;Src- DOI 10.5281/zenodo.3475601 | Schaperow et al., 2021 |
| 8 | EGM2008 | Geoid model that provides mean sea level in meters everywhere on Earth. | Resolution – 5 arc minutes; Src- National Geospatial-Intelligence Agency (NGA) | Pavlis et al., 2012 |

**Table 3: Different datasets along with their description and source, that are included in the global database and are used as default inputs by RAT 3.0 if downloaded by the user.**

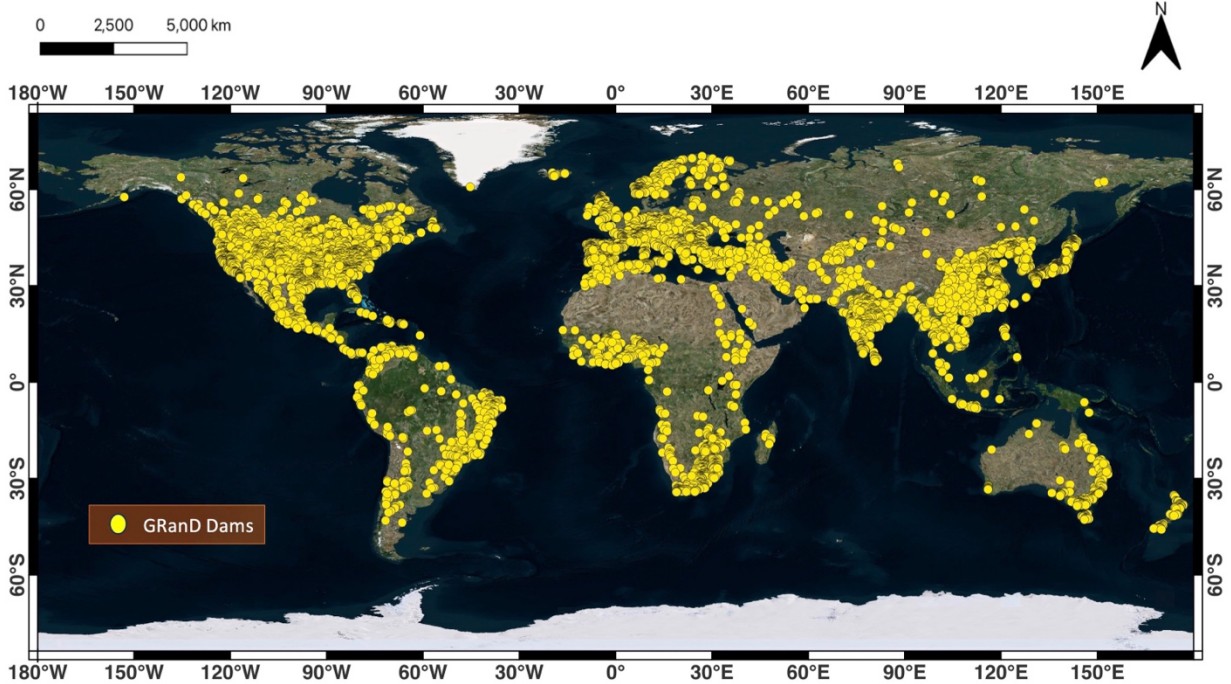

**Figure 1: Geographic representation of major dams around the world derived from a global dam database known as GranD (Lehner et al., 2011)**

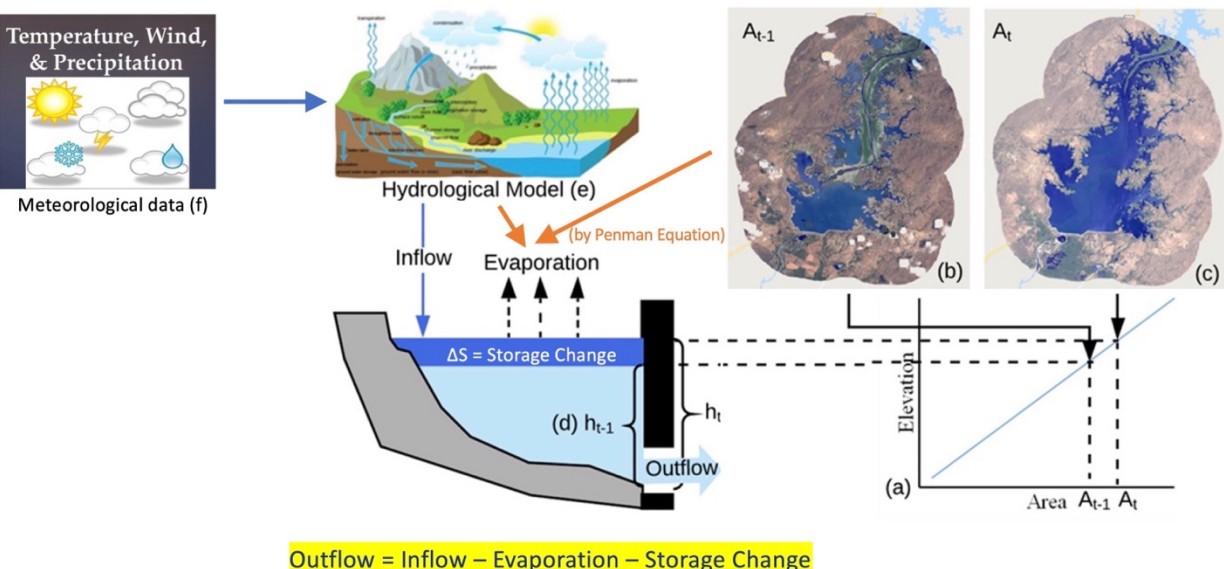

**Figure 2: Conceptual model of RAT and the mass balance equation.**

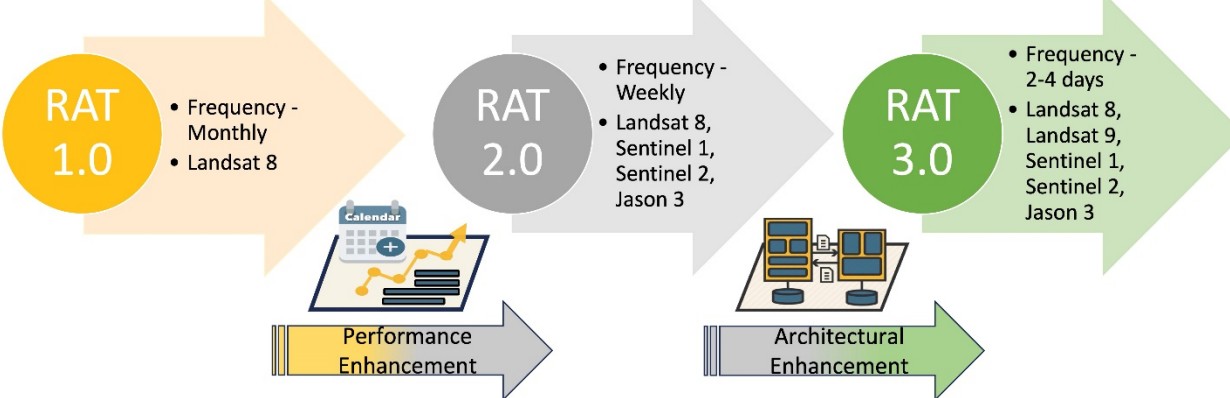


**Figure 3. Technological evolution of the RAT software from version 1.0 to version 3.0. Names of key satellite missions are mentioned in the second bullet of each version.**

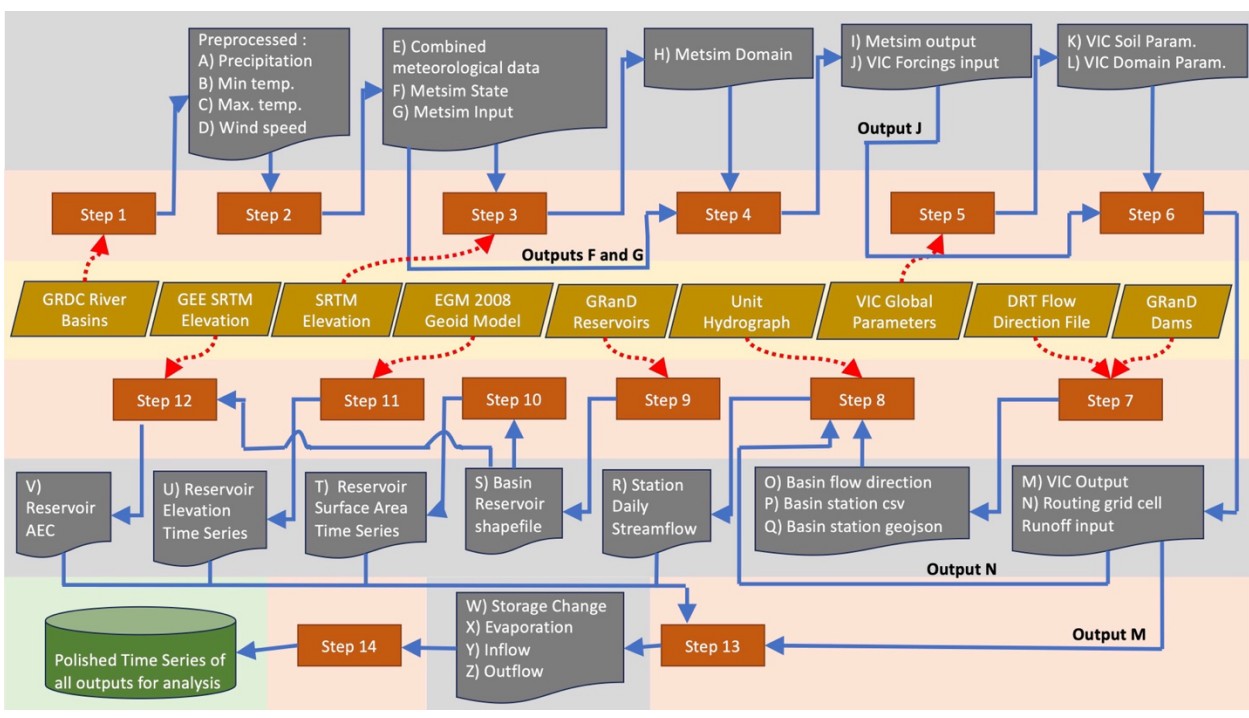


**Figure 4: Workflow of RAT 3.0 with all the different steps listed in Table 1. The yellow layer shows the user inputs, the orange layer shows the flow of different steps or processes, and the grey and green layers show the intermediate and final outputs produced by RAT 3.0, respectively.**

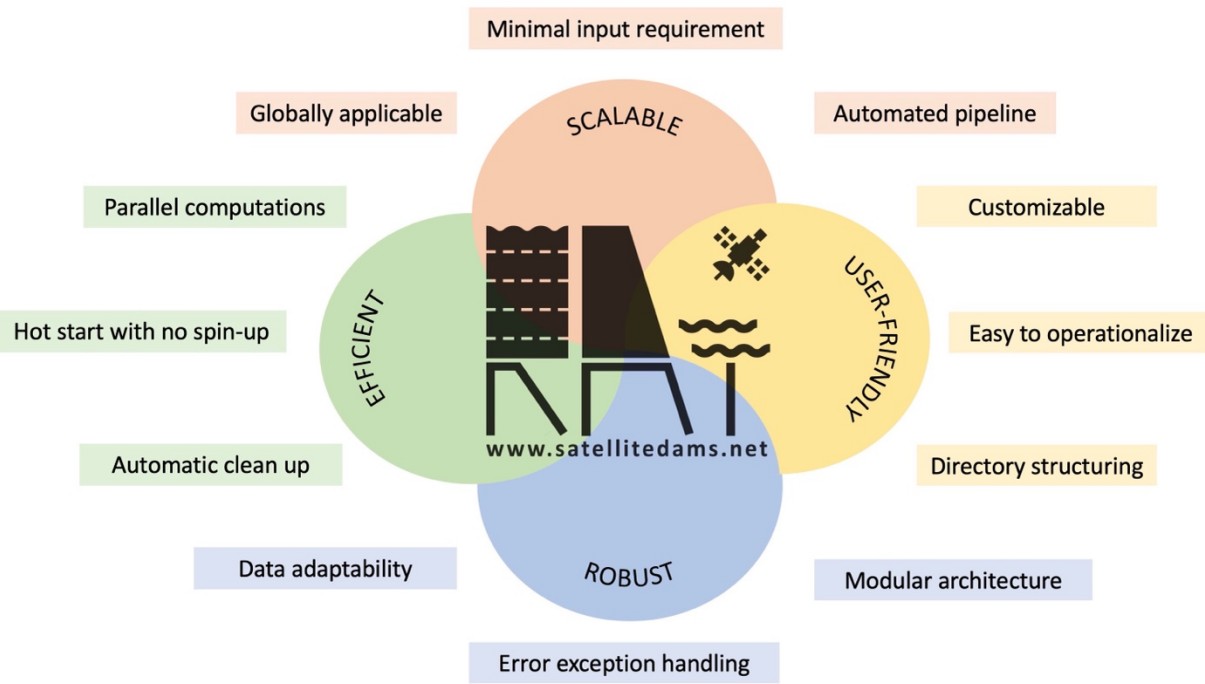

**Figure 5: Major software design advancements in RAT 3.0.**

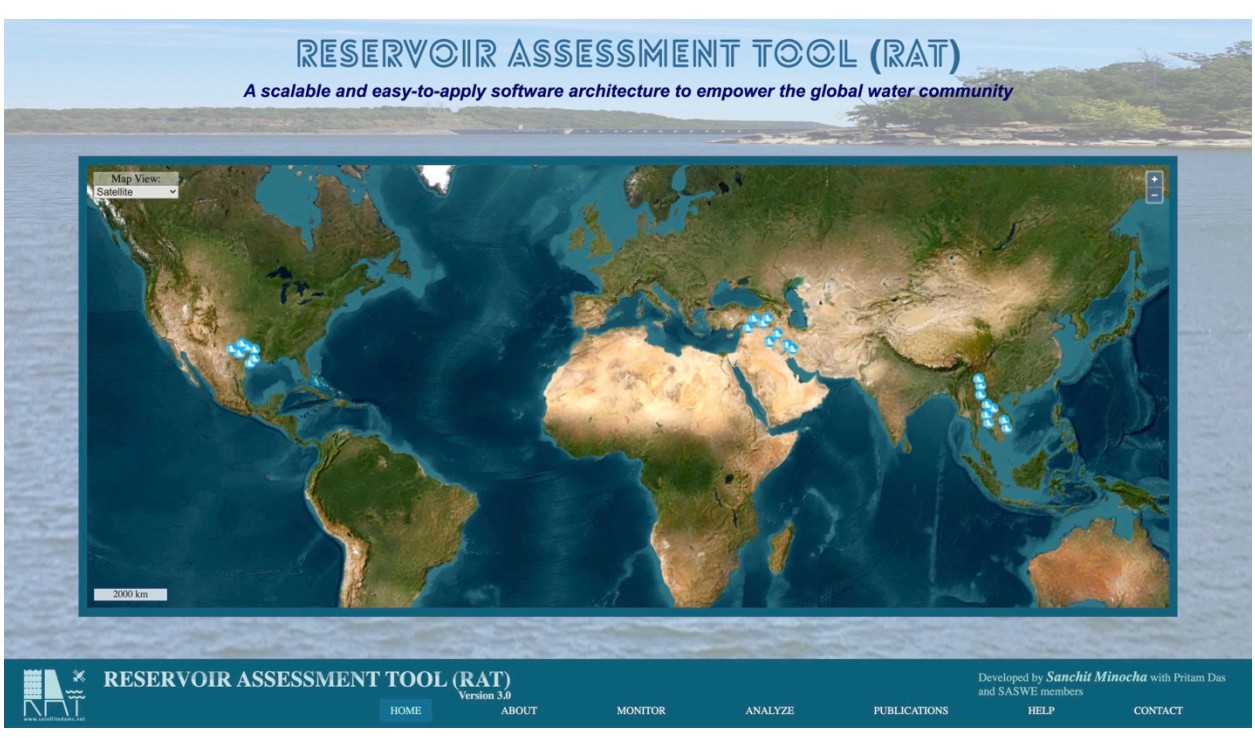

**Figure 6: Homepage of the front end of the global RAT 3.0's online set up at www.satellitedams.net  On the HELP menu, the RAT 3.0 software can be downloaded as well as other resources such as training and education can be found.**

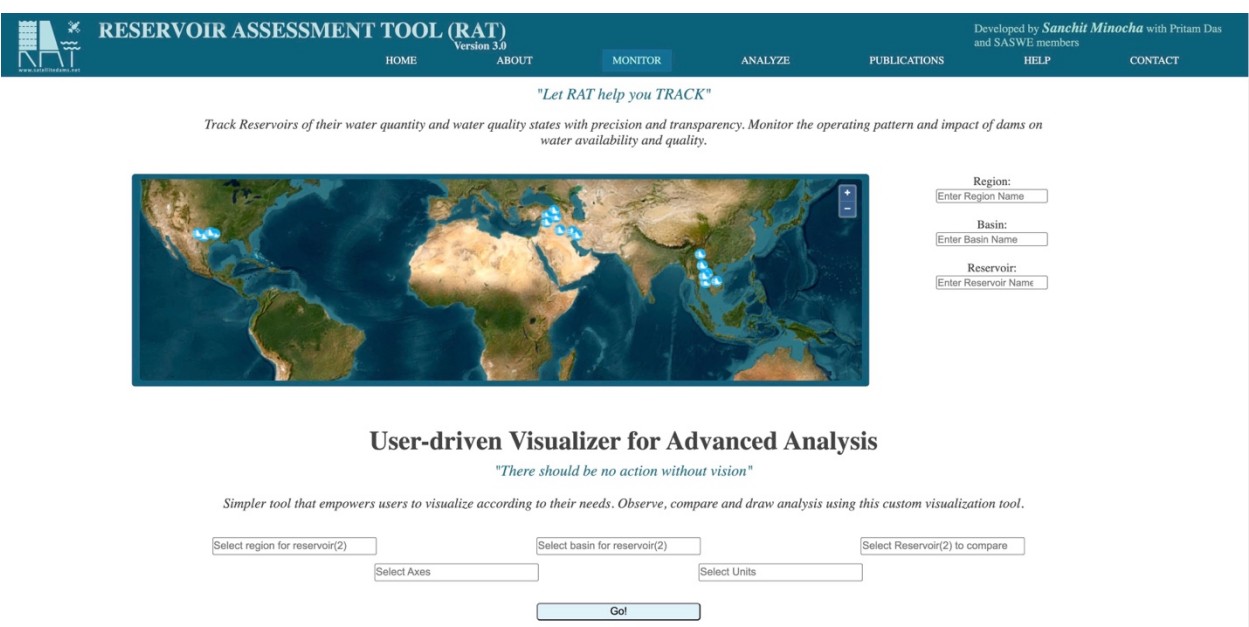

Figure 7: Advanced visualizer feature to meet the diverse requirements of different entities and stakeholders.

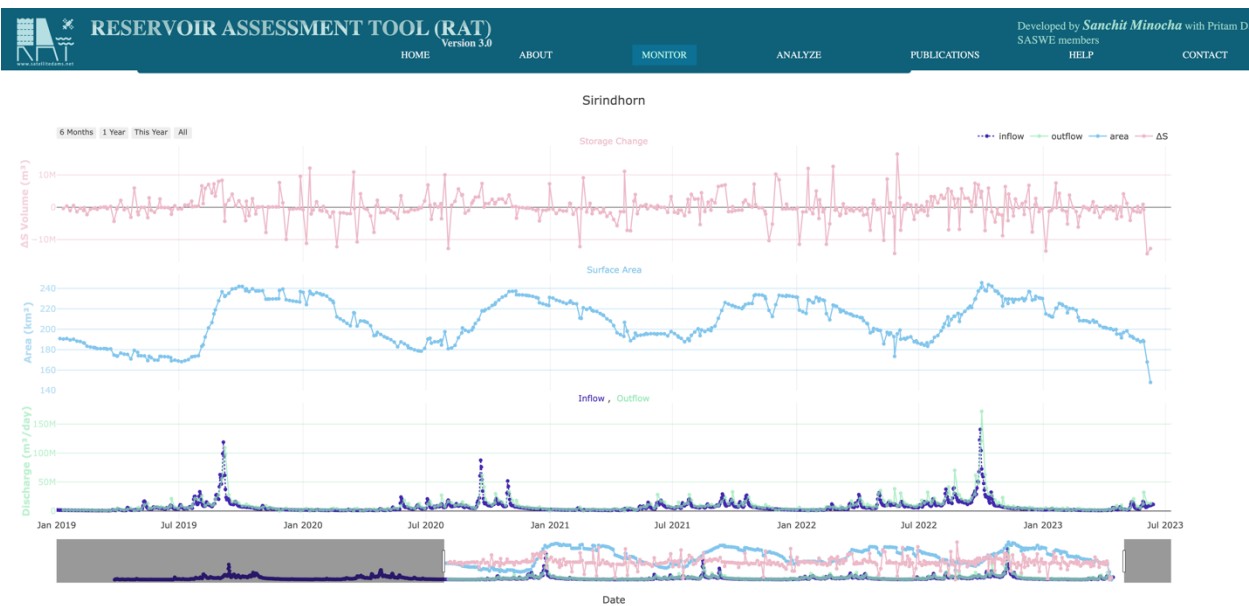

**Figure 8: Reservoir data for one of the reservoirs 'Sirindhorn' (shown in Figure 6 (e)) in the Mekong River basin produced by RAT 3.0.**



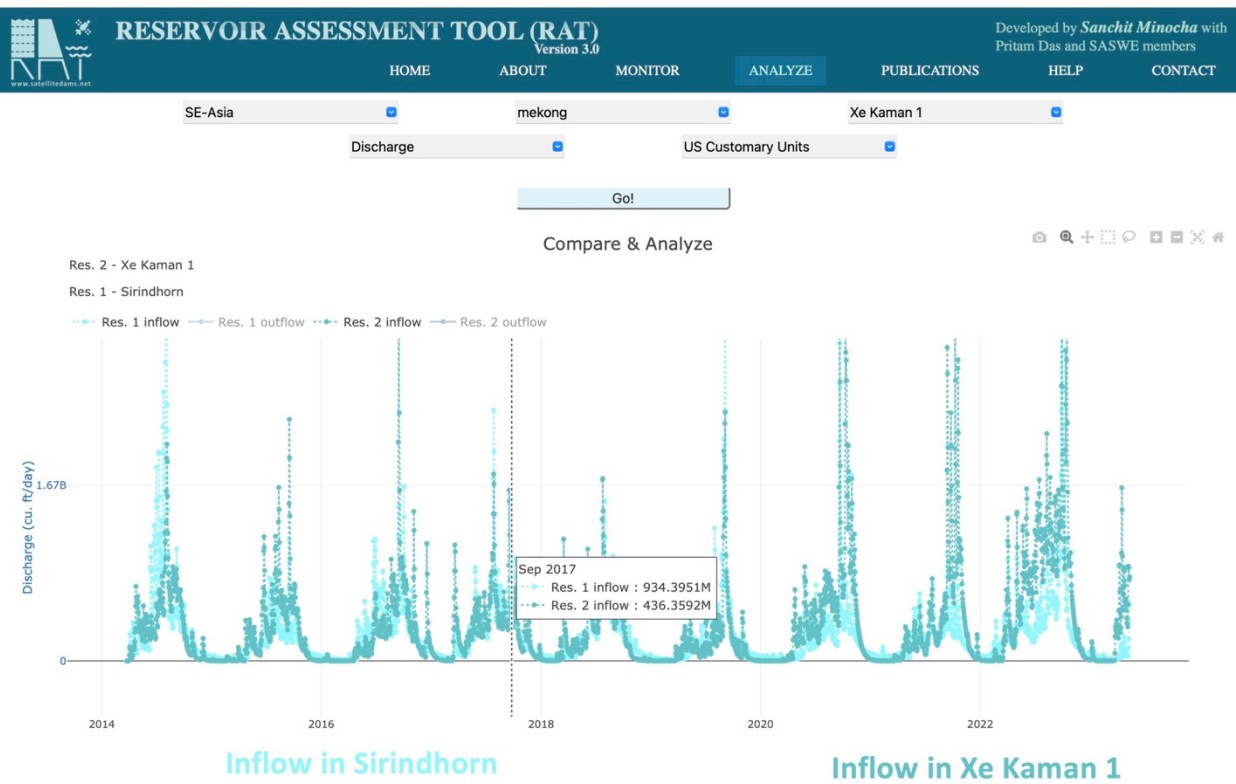

**Figure 9: Inflow for 'Sirindhorn' in the Mekong River basin is compared to another nearby reservoir 'Xe Kaman 1' (shown in Figure 6 (e)) using the advanced visualizer feature.**

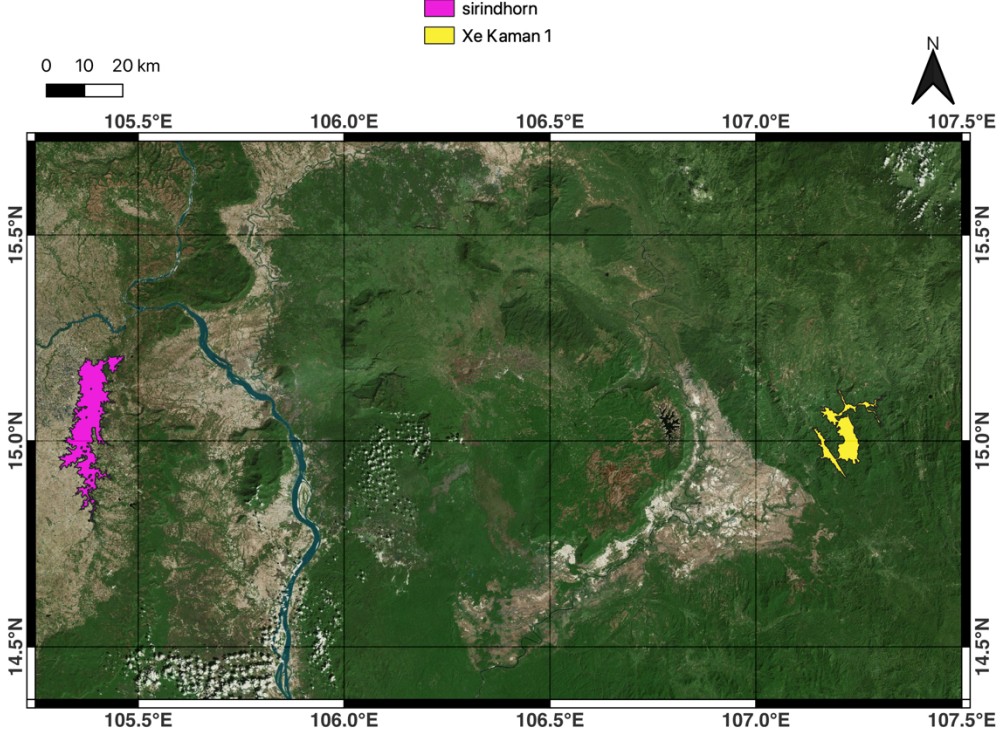


**Figure 10: Reservoirs Sirindhorn and Xe Kaman 1 shown on the map for comparative analyses of RAT outputs on the global web app of RAT at www.satellitedams.net.**

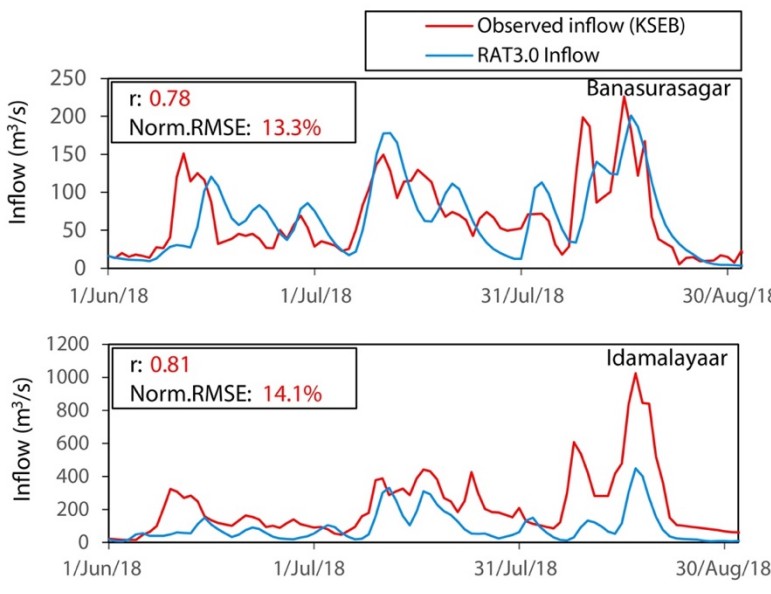

**Figure 11: RAT hydrographs comparing observed inflow at Banasurasagar and Idamalayaar dam (red) against hydrologic model inflow by RAT using VIC (blue) during the 2018 Kerala floods. After Suresh et al. (2023). See Fig 12 for a map of Banasurasagar and Idamalayaar dam in Kerala.**


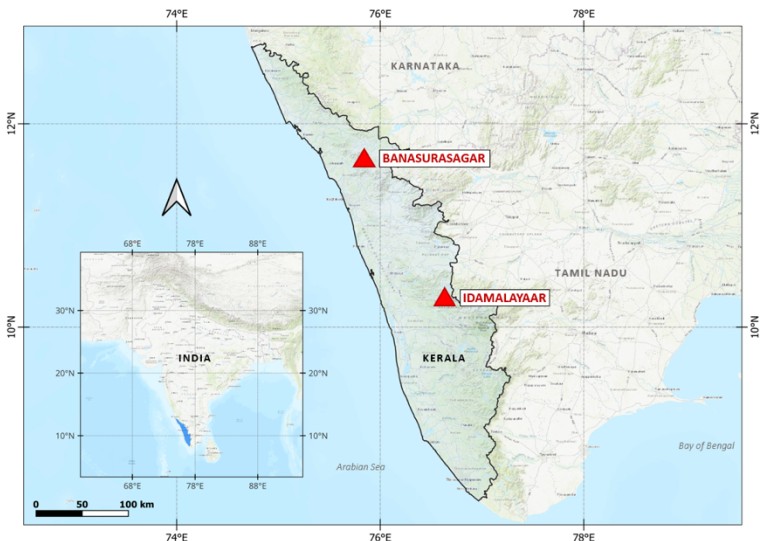

Figure 12: Map of Kerala with the location of the Banasurasagar and Idamalayaar dam. After *Suresh et al. (2023)*

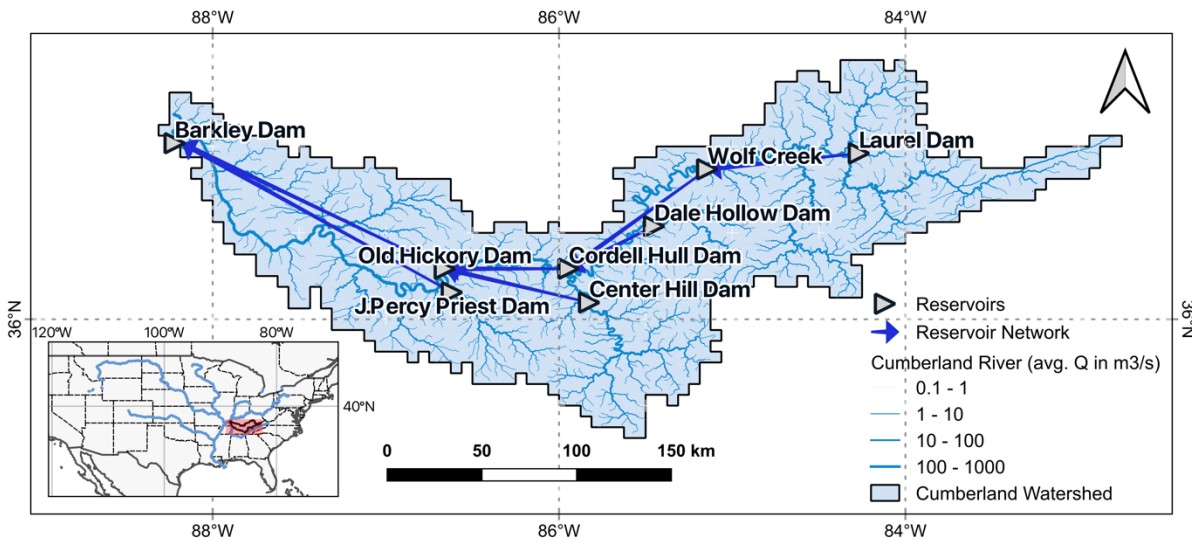

Figure 13: Map of the Cumberland basin where the ResORR algorithm to model regulated inflow for RAT was developed and test. The map shows locations of the reservoirs, the reservoir network, and the location of the Cumberland basin in the US. After *Das et al. (2023)*.

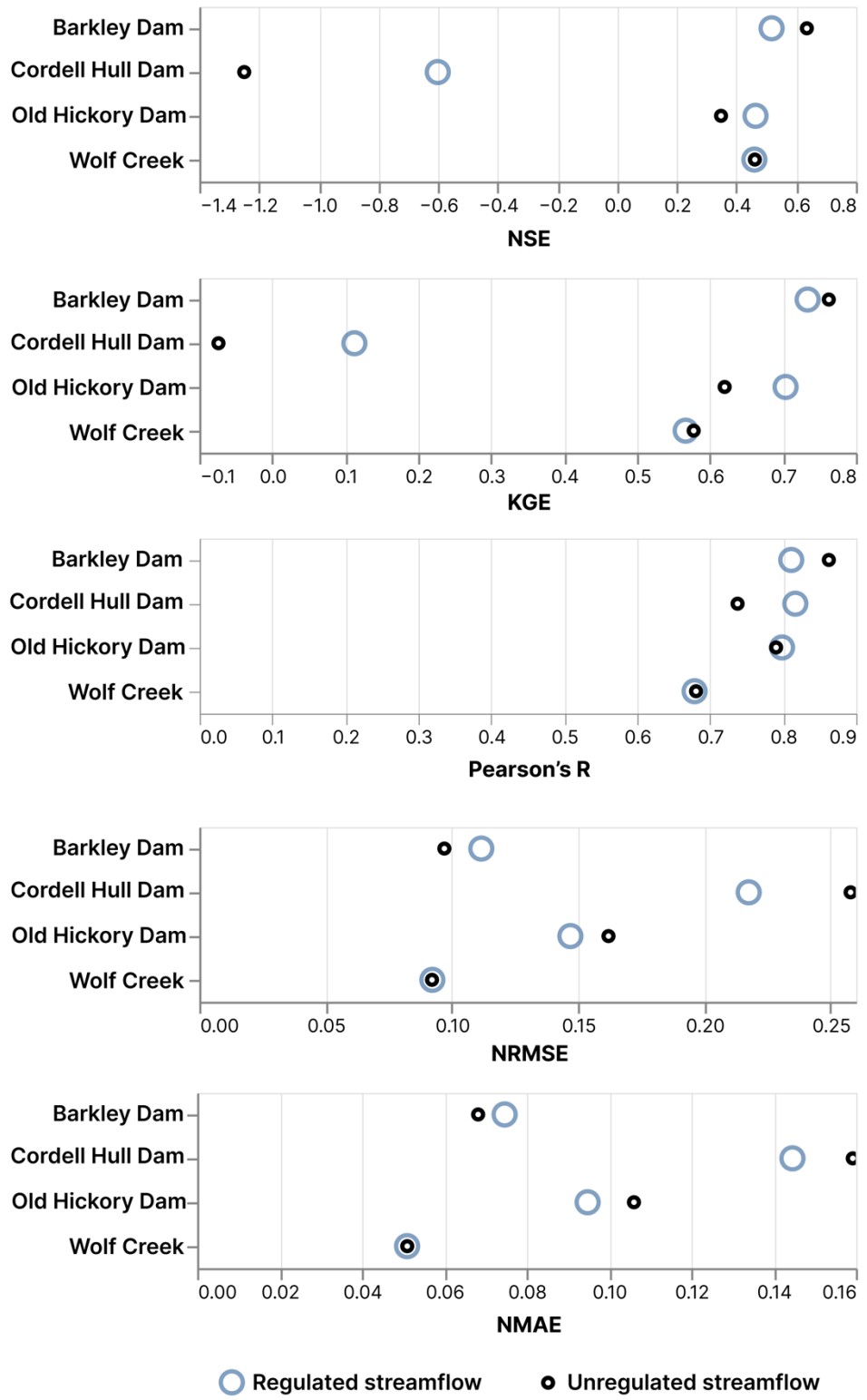

**Figure 14: Performance of RAT-generated regulated inflow using ResORR compared to the unregulated inflow modeled using VIC in the regulated Cumberland River basin in terms of performance metrics. The chart shows that it is clearly improving simulations when compared to the baseline (unregulated inflow) for most dams. After *Das et al. (2023)*.**

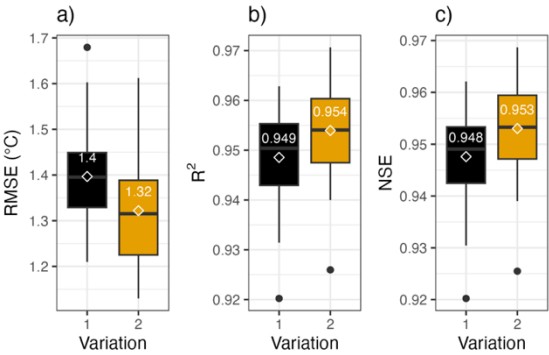

**Figure 15: The impact of using RAT based reservoir storage change on improving river temperature prediction. Black is without using RAT and yellow is with using RAT storage change at more than 50 river reaches downstream of dams in Columbia river basin. After *Darkwah et al. (2023).***

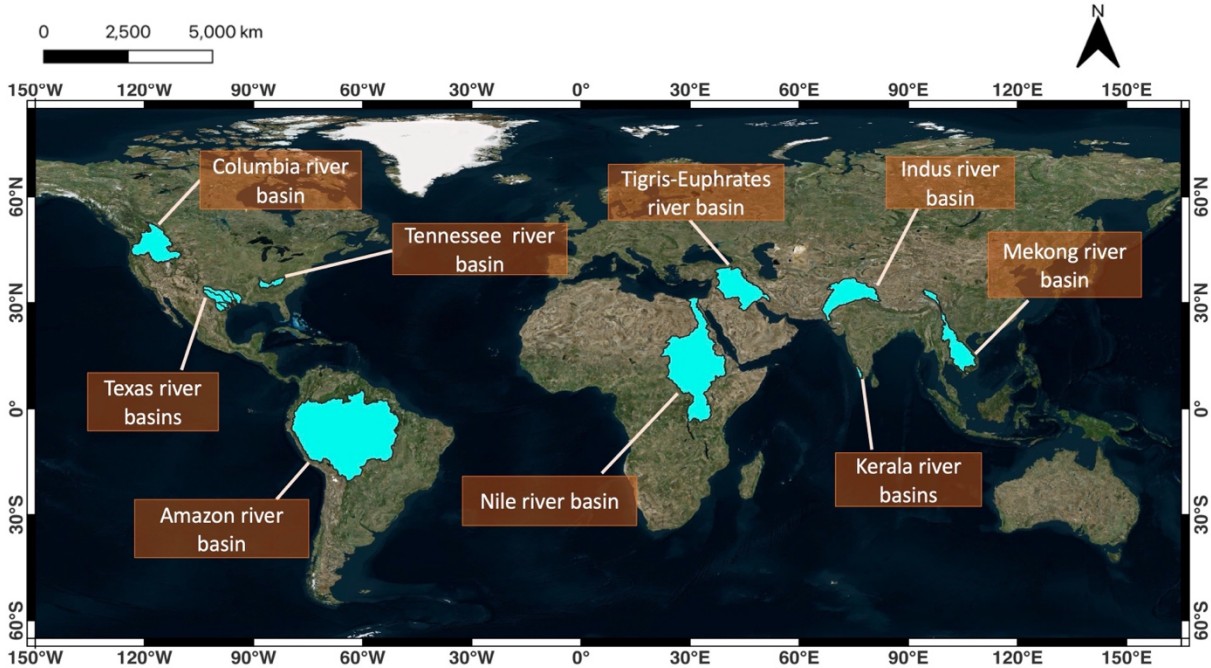

**Figure 16: River basins where RAT 3.0 has been (or is being) independently installed and tested by users.**