# Peer review of "Reservoir Assessment Tool Version 3.0: A Scalable and User-Friendly Software Platform to Mobilize the Global Water Management Community"

_Geoscientific Model Development, 2023_

## Referee Comment (RC2)

[referee-annotated manuscript omitted]

---

## Author Comment (AC1)

**POINT BY POINT RESPONSE TO REVIEWERS**

We appreciate the editor's help in providing us with timely reviews for our manuscript. We also appreciate time and effort the reviewers spent in helping us to improve the quality of our manuscript. We have considered each comment very seriously and performed additional and extensive analyses where appropriate to improve the quality of our manuscript. Below we summarize the key additional work we have undertaken to address the reviewer's concerns:

- ❖ We have now included the scientific motivation and questions that RAT can answer in the manuscript.
- ❖ We have added an additional table that describes the advancements in RAT 3.0 as compared to RAT 2.0.
- ❖ We have provided validation against in-situ data
- ❖ To provide better idea of the global database for readers, we have incorporated additional information for various datasets in the global database.

In the section below, our response to each reviewer comment is shown in blue while the reviewer's comments are in black.

**SPECIFIC RESPONSES**

**Clarify Scientific Questions and Value:** The manuscript focuses heavily on technical aspects. It needs to build a stronger scientific motivation. Articulate the specific scientific questions or knowledge gaps that RAT 3.0 addresses and discuss how the estimated reservoir parameters advance the understanding of reservoir hydrology or operations. Include scientific findings and insights from RAT 3.0 applications. There needs to be more discussion and quantification of the scientific value of the reservoir parameters estimated by RAT 3.0. How do they advance understanding of reservoir hydrology or operations? Any new scientific findings or hydrologic insights gained from sample RAT 3.0 applications should be included to demonstrate its scientific utility.

**Response:** Thanks for the comment and bringing our attention to this aspect of the manuscript. We agree with the reviewer that we need to discuss the knowledge gaps that RAT 3.0 has the potential to answer. Therefore, in response to the reviewer's constructive feedback regarding the need for a more robust scientific motivation in the manuscript, we have refined the discussion to better articulate the scientific questions addressed by RAT 3.0 and the value it can bring to reservoir hydrology operations. Here is what has been added to the manuscript in section 2.2 (in quotes).

[revised manuscript text omitted]

2. Use of RAT 3.0 to assess the impact of the Belo Monte Dam on the Xingu River in the Amazon (Hossain et al., 2023). In this study, RAT 3.0 was able to pick up the historical hydrologic response to precipitation of the region during the post-dam period (after the Belo Monte dam was constructed).

3. Use of RAT 3.0 to develop algorithm to estimate regulated inflow using VIC's natural inflow (which assumes the absence of upstream dams) with study site as Tennessee River

basin (Das et al., 2023). In this study RAT 3.0 with modeling of regulation of flow by upstream reservoir storage was found able to consistently predict well the downstream inflow to a series of reservoirs (see Figs 3 and 4 below).

[Figure]

**Fig 3:** Map of the Cumberland basin where the ResORR algorithm to model regulated inflow for RAT was developed and test. The map shows locations of the reservoirs, the reservoir network and the location of the Cumberland basin in the US. *After Das et al. (2023)*

[Figure]

**Fig 4.** Performance of RAT-generated inflow to the downstream reservoir in the regulated Cumberland River basin in terms of performance metrics. The satellite-based delta S is derived from RAT and shows that it is clearly improving simulations when compared to the baseline (called TNR – Theoretical Natural Runoff) for most dams. After Das et al. (2023)

4. Valuable insights provided by RAT into the effects of dams in the transboundary river basin of Mesopotamia (Hossain et al., 2023). In this study, RAT was developed, set up and operationalized over the Tigris Euphrates river basins exclusively to demonstrate to stakeholders such as Iraqi Ministry of Water Resources what is possible for transboundary reservoir monitoring using satellite data alone. Readers can see the operational system running at http://depts.washington.edu/saswe/rat The scientific potential of this RAT rendition is that it brought various communities together to discuss how the Mesopotamian rivers could be restored and how the downstream marshland could be revised using techniques such as RAT when there is no in-situ data or publicly available modeling tools.

5. Analyzing the role of reservoirs on downstream river temperature using Rat 3.0 for Columbia River basin (Darkwah et al. 2023). In this study, RAT generated reservoir storage was able to improve the remote sensing based surface water temperature estimation in the 50 km downstream reach of the Columbia river dams. With RAT derived storage change, there was about 50% reduction in RMSE of temperature prediction .

[Figure]

**Fig 5.** The impact of using RAT based reservoir storage change on improving river temperature prediction. Black is without using RAT and yellow is with using RAT storage change at more than 50 river reaches downstream of dams in Columbia river basin. After *Darkwah et al. (2023)*

**Detailed Comparison of RAT 2.0 and 3.0**: To provide a comprehensive understanding of the advancements, I recommend including a detailed comparison between RAT 2.0 and 3.0.

**Response:** Thank you for your comment. We acknowledge that providing a detailed comparison between RAT 2.0 and RAT 3.0 could enhance the reader's understanding of the advancements. However, we are mindful that such a comparison might shift the reader's focus from appreciating the inherent user-friendly power and user-centric capabilities of RAT 3.0 to merely assessing its quantitative accuracy over RAT 2.0. While we believe it is implicit that the advancements in RAT 3.0 surpass those in RAT 2.0, we agree that offering a brief and focused overview of the

differences between the two versions could be beneficial. Consequently, we have incorporated a concise table in the paper that succinctly compares RAT 2.0 with RAT 3.0.

For your reference, here is a detailed comparison between RAT 2.0 and RAT 3.0:

1. **Scalability**:
   a) RAT 2.0 was a set of code that was hard coded for the dams in the river basin of Mekong but RAT 3.0 has been generalized and is applicable for any reservoir worldwide.
   b) Apart from the hard coded scripts, the input requirements in case of RAT 2.0 were a lot which further makes it difficult to apply the same for dam of any other river basin. RAT 3.0 effectively handles the automatic generation of a lot of these inputs, therefore significantly reducing the number of inputs required.
   c) In RAT 2.0, there was a disconnect between code of hydrological modeling and the code of google earth engine requiring the user to create assets first in the user's account. Also, the automatic input generations were not there. This has led to a lot of manual work for the user. In RAT 3.0, all of this is handled completely automatically. User does not need to create assets in google earth engine account and all the inputs are generated automatically using global and publicly available datasets.

2. **Robust**
   a) RAT 2.0 and RAT 3.0 both uses real-time meteorological datasets from different servers. RAT 3.0 handles the missing data in a better way as compared to the RAT 2.0. For example if a temperature data of a single day has been missing, it will automatically be interpolated in time domain.
   b) Error and exception handling has been significantly improved for RAT 3.0 as compared to RAT 2.0. In Rat 2.0, only a single log file was created which was difficult for a user to follow. In RAT 3.0, this log file is accompanied by another short and summarized log file which can be easily followed by a user.
   c) In RAT 2.0 if the hydrological model fails for some reason, the rest of the code won't be executed. In RAT 3.0, if some part of a code related to hydrological modeling fails to generate inflow, the earth engine code will still run for surface area computations. This is because of the modular architecture of RAT 3.0.

3. **User-Friendliness**
   a) In RAT 2.0, the outputs produced by RAT were not organized into proper directories making it difficult for a user to track down the different outputs given that the number of output files generated by RAT are a lot. In RAT 3.0, the directory structure has been reorganized in an intuitive manner so that user can easily track down outputs produced at different stages of RAT while execution.
   b) For RAT 2.0, maximum of all inputs was handled using a configuration file but there were some inputs like user has to upload assets to one's earth engine account. RAT 3.0 handles all the inputs using a single configuration file. Though the input requirements are minimal for beginners, advanced users can easily customize the inputs according to their needs using the configuration file.
   c) RAT 3.0 has an inbuilt feature that a user can use to operationalize it using a cron job. For RAT 2.0, a user has to set a different cron job of manually changing configuration file every time before executing RAT in an operational manner.

4. **Efficiency**
   a) RAT 2.0 executed the hydrological model from the very beginning even for operational jobs which is less efficient both time and resource wise. RAT 3.0 now provides a feature to hot-start from where the execution stopped the last time. It saves the last state of the model to do so.
   b) A lot of processes in RAT 3.0 have been parallelized using dask python library. Downloading of meteorological data and execution of routing model for different dams are some jobs which make use of the parallel computations. RAT 2.0 was not using the implementation of parallel processing.
   c) The extensive set of output files generated by RAT 2.0 used to pile up, taking a lot of memory space making RAT memory inefficient. RAT 3.0 provides a feature to the user to select automatic deletion of intermediate output files for its different modules.

**Validation Against In-Situ Data:** The manuscript would greatly benefit from an expanded section on validation. A quantitative skill assessment comparing RAT 3.0 outputs to in-situ data will enhance the credibility and reliability of the tool.

**Response:** We appreciate the reviewer's insightful feedback and acknowledge the importance of validation in enhancing the manuscript's robustness. We have already addressed this comment in one of our earlier responses above where we summarized the five instances of quantitative scientific validation of RAT 3.0.

**Elaboration on the Global Database:** To provide readers with a clearer picture, please elaborate on the sources, coverage, resolution, and update frequency of the global database utilized in RAT 3.0. This will help users assess the data's suitability for their specific applications.

**Response:** Thank you for your valuable suggestion. We have included additional details for global database as shown in below Table. This table provides insights into the source, version, resolution, and coverage of each dataset, as applicable. This expanded information aims to better assist readers in evaluating the suitability of the database for their specific applications.

| S. No. | Dataset Name | Brief Description | Additional Information | Reference |
|--------|--------------|------------------|------------------------|-----------|
| 1 | GRDC Major River Basins | Shapefile of all the major river basins' polygons in the world. | Version - 2nd Revised Edition, 2020; Src- The Global Runoff Data Centre | GRDC, 2020 |
| 2 | GRanD Dams | Shapefile of all the major dams' locations in the world. | Version - 1.3 ; Src - https://www.globaldamwatch.org | Lehner et al., 2011 |
| 3 | GRanD Reservoirs | Shapefile of all the reservoirs' polygons associated with dams in [2]. | Version - 1.3 ; Src - https://www.globaldamwatch.org | Lehner et al., 2011 |
| 4 | DRT Flow Direction | Global raster file providing the flow directions in WGS84 projection. | Resolution - 1/16º; Src- Numerical Terradynamic Simulation Group , University of Montana | Wu et al., 2011, 2012 |
| 5 | SRTM_30 Elevation | Global elevation raster file providing elevation in meters. | Resolution – 30 arc seconds ; Src - Scripps Institution Of Oceanography, University of California San Diego | Becker et al., 2009 |

| 6 | VIC Soil Parameters | Global VIC soil parameter dataset containing separate raster files for each continent in WGS84 projection. | Resolution - 1/16º; Spatial coverage from 60°S to 85°N; Version-1.6d ;Src- DOI 10.5281/zenodo.3475601 | Schaperow et al., 2021 |
|---|---|---|---|---|
| 7 | VIC Domain Parameters | Global VIC domain parameter dataset containing separate raster files for each continent in WGS84 projection. | Resolution - 1/16º; Spatial coverage from 60°S to 85°N; Version-1.6d ;Src-DOI 10.5281/zenodo.3475601 | Schaperow et al., 2021 |
| 8 | EGM2008 | Geoid model that provides mean sea level in meters everywhere on Earth. | Resolution – 5 arc minutes; Src-National Geospatial-Intelligence Agency (NGA) | Pavlis et al., 2012 |

---

## Author Comment (AC2)

**POINT BY POINT RESPONSE TO REVIEWER 2**

We appreciate the editor's help in providing us with timely reviews for our manuscript. We also appreciate time and effort the reviewer spent in helping us to improve the quality of our manuscript. We have considered each comment very seriously. Below we summarize the key additional work we have undertaken to address the reviewer's concerns:

- ❖ We have edited the manuscript to include the annotated suggestions by the reviewer.
- ❖ We have acknowledged the limitation of RAT in accounting for sedimentation of reservoirs in the manuscript and mentioned that SWOT satellite mission will be able to solve the problem.
- ❖ We have mentioned extent of retrospective analysis afforded by historical satellite data and how far back one can derive reservoir operations data using RAT.
- ❖ We have discussed the role of hydrologic model calibration in RAT in the context of model agnosticism.
- ❖ We have clarified the usage of the term 'user-friendly' in the manuscript wherever applicable.
- ❖ The system requirements for RAT 3.0 have now been mentioned in the manuscript.

In the section below, our response to each reviewer comment is shown in blue while the reviewer's comments are in black.

**SPECIFIC RESPONSES**

Sedimentation, especially in medium to small reservoirs can impact storage capacity significantly and often is not considered when designing a dam because of the cost associated to implement mitigative measures. While the authors mention sedimentation as an issue, nothing is said about if/how RAT 3.0 might deal with changing reservoir storage or its limitations. I believe a little more discussion/reference to and impact it has on RAT 3.0 would be helpful.

**Response:** This is a good point. Sedimentation into a reservoir, regardless of its size, modifies the area-elevation relationship (the hypsometric curve) that defines the bathymetry. Thus, conceptually, RAT 3.0 can handle sedimentation as the area-elevation data is easy to modify or update. User can define their bathymetry data and such bathymetry data can be swapped, updated or modified in a modular fashion. In general, RAT yields the highest accuracy when the bathymetry is based on in-situ topographic surveys pertaining to the period of reservoir simulation. Such modularity in area elevation relationship of reservoirs has been taken advantage of in RAT applications in the Mekong river basin (see Das et al., 2022) and in the Cumberland river basin (see Das et al., 2023).

Despite this modularity in representing bathymetry in RAT 3.0, there is an inherent limitation in global scalability of addressing sedimentation issues. This is because the global digital elevation models (DEM) that exist today are mostly from satellite remote sensing (such as from Shuttle Radar Topography Mission – SRTM) and lack temporal variability. For example SRTM DEM captures the reservoir bathymetry for all reservoirs large or small of the world that were built before 2000 and from the water level that existed during sampling. While such a global DEM

provides the complete bathymetry of reservoirs created after sampling time (for SRTM it is February 2000), it does not represent the sedimentation that has taken place since then. For pre-2000 reservoirs, the SRTM DEM is limited due to the submerged part of the DEM requiring parameterizations or additional satellite observations when reservoir was at lower levels  (see Bonnema et al. 2017). Thus, the ability of RAT 3.0 to address sedimentation at the global or regional scale is currently limited primarily due to the lack of time-varying global DEM data. Currently overcoming this limitation is an active area of research for RAT 3.0 using sediment trapping concepts and satellite observations of sediment concentration.

In the manuscript, we plan to add the following the Discussion and Conclusion section as follows to address the reviewer's comment:

"The impact of sedimentation on reservoir storage is a critical consideration addressed in RAT 3.0. The framework's modular design allows users to dynamically update the area-elevation relationship, ensuring accurate simulations even in the presence of sedimentation. Localized applications in the Mekong and Cumberland River basins have showcased the framework's adaptability in representing changes in bathymetry due to sedimentation (Das et al., 2022; Das et al., 2023). However, extending this capability globally faces challenges due to the lack of time-varying global digital elevation models (DEM). For pre-2000 reservoirs, such as those covered by the Shuttle Radar Topography Mission (SRTM), representing sedimentation requires additional parameterizations or satellite observations when the reservoir was at lower levels (Bonnema et al., 2017). Future research endeavors with RAT 3.0 will explore potential solutions, involving sediment trapping concepts and the integration of satellite geodetic observations, to enhance its global scalability in addressing sedimentation challenges."

Over what time is the model run? I don't understand if I can run this for a few months, a few years, or back into the 1900's (to what date does satellite data support reservoir surface calculations?). Maybe this is addressed in papers discussing v1 or v2, but I would like to know some recommendations/limitations on specifying data ranges and run time lengths?

**Response:** Thank you for your comment and bringing attention to the temporal scope of the model. The timestep of RAT's tracking of reservoir state is controlled by the time step of the hydrologic model and frequency of satellite observation. Currently, RAT 3.0 is able to run for a period as small as a single day to as long as multiple decades. In its current formulation, RAT 3.0 yields the best performance when multiple area-observing satellites in the microwave and visible wavelengths are synergized in the reservoir storage algorithm called Tiered Multi-sensor-Optical/SAR (TMS-OS; Das et al., 2022). Using Sentinel-1 SAR (microwave) and Sentinel-2 and Landsat satellites in the visible wavelength, RAT can be reliably operated from 2015 onwards. However, RAT 3.0 can also be run from the time of earliest availability of visible satellites from the Landsat 4 mission in 1985. This means that technically RAT3.0 can be run for more than 3 decades starting from the early 80s. In fact, in a recent application of RAT 3.0 for predicting river temperature using satellite data, RAT 3.0 was run from 2000 using Landsat data only.  We have added an appropriate summary of this issue in the manuscript in the Discussion and Conclusion as follows:

"RAT 3.0 simulation can span from a single day to multiple decades, and is dictated by the hydrologic model's smallest time step and satellite observation frequency. The Tiered Multi-sensor- Optical/SAR (TMS-OS) algorithm, that calculates reservoir storage change based Sentinel-1 SAR, Sentinel-2, and Landsat satellites, allows RAT 3.0 to operate from 2015 onwards. However, single-sensor functionality can be extended to the early 80s, utilizing Landsat 4 data as has been demonstrated by Biswas and Hossain (2021). "

How much confidence is there in the calculated hydrology of VIC. What if any calibration has been done? I have done enough hydrologic modeling locally and globally with GEOGloWS to know that this is not a simple matter, but the lack of discussion about how good you can expect the inflows computed from the VIC model with global data is a shortcoming of the manuscript. Again, maybe this is given more in other papers, but some further discussion about the quality of the hydrologic model would strengthen the paper. As a side note I'm interested in seeing how GEOGloWS or other models might substitute for the hydrologic inputs to the reservoir as the developers make that possible with the minor upgrades they mentioned. I beleive that will make an important difference to the community of users who often have significant investments in their own hydrologic models (like the National Water Model in the US and EFAS in Europe along with GEOGloWS and other global models like GloFAS).

**Response:** This is a great point. As RAT is dependent on hydrologic model for inflow, calibration and the choice of the hydrologic model are important topics. The performance of RAT will depend on how well calibrated and representative the gridded hydrologic model is in capturing the upstream most reservoir inflow that is not regulated (i.e. at the head waters). In a recent study by Das et al. (2023) to account for upstream reservoir regulation in predicting the downstream reservoir inflow and RAT-based outflow for lower reservoirs, it was reported that calibration of the hydrologic model is important mostly up to the last upstream boundary point where flow remains naturalized. Beyond that point, calibration ceases to become very sensitive as the assimilation of reservoir storage change (hedging or release) to account for regulated flow downstream using an appropriate regulation algorithm becomes more important (see Das et al. 2023). We plan to demonstrate the impact of hydrologic model calibration and application of flow regulation at multiple river basins around the world in a follow up paper under preparation for the journal Environmental Modeling and Software.

We are also heartened by the reviewer's enthusiasm to alternate model-based streamflow simulations in RAT to account for reservoir regulation. We believe this is quite feasible in the current RAT 3.0 architecture as the current hydrologic model (VIC 5.0) to predict streamflow can be made architecturally modular from the satellite data processing component to estimate reservoir storage change and outflow. Thus, alternate flow data such as GEOGLOWS and forecast flow from GloFAS can replace the current VIC model in the RAT 3.0 architecture, pending appropriate modifications and certain data format, structure and error handling instructions are maintained.

We have addressed the reviewer's comment by discussing this in the Discussion and Conclusion section of the paper as follows:

"The calibration and choice of the hydrologic model play a crucial role in RAT's performance, as it relies on the accuracy of unregulated inflow capture, especially at the headwaters. A recent study (Das et al., 2023) highlights the importance of hydrologic model calibration up to the last upstream boundary point where flow remains naturalized. Beyond this point, sensitivity to calibration decreases, emphasizing the significance of assimilating reservoir storage changes to account for regulated flow downstream using an appropriate regulation algorithm."

Also, additional information regarding the VIC Global Parameters has been added to the manuscript in the section 3.5 that includes how calibration of these parameters is required to get better results.

"Please note that the VIC Global parameters are uncalibrated. Therefore, users are required to calibrate intricate yet challenging-to-measure parameters, including soil depth and the variable infiltration capacity parameter. Achieving an accurate match between simulated and observed inflow requires careful calibration efforts. For a comprehensive understanding of the calibration process and additional details, we recommend referring to the Usage Notes provided by Schaperow et al. (2021)."

Do you have documented use cases of others implementing RAT 3.0? How do you know it is user-friendly otherwise? Other than Figure 11 which showed where RAT 3.0 was implemented, I don't really see evidence of the user-friendliness which even the title uses for local implementations, especially "farmers" which it is implied could use it. I believe farmers could benefit, but not implement and maybe that was the intention here, but even discussing other implementers such as local dam or water districts are not discussed. In some places the manuscript speaks of open software and a developer community and I believe the user-friendliness applies to this group of people (developers) but not as much to stakeholders and sometimes that language in the paper treats both as the same. I think it would be better to distinguish that in the paper.

**Response:** Thank you for your feedback. We appreciate your attention to the clarity of the term "user-friendly" and its application to different user groups. In response to your feedback, we would like to elaborate on the user-friendliness of RAT 3.0, particularly for diverse user communities.

In the current history of RAT's development, version 1.0 was developed and published in 2021, followed by version 2.0 in 2022 and finally version 3.0 in 2023. It is only in version 3.0 that the RAT can be applied for any river basin across the world in the so-called user-friendly manner as it is available with the most comprehensive user documentation and tutorials. Because the user friendly version of RAT 3.0 is very recent, we need more time to observe if users around the world are indeed able to independently set RAT 3.0 and if the barrier to entry has indeed been lowered. We are heartened by two facts on why we believe RAT 3.0 is more user-friendly than the current state of the art (at least compared to version 1.0 or 2.0). First – at the time of writing this revision, RAT 3.0 software has been downloaded around the world more than 1100 times in 3 months with the lionshare (> 75%) of downloads being from outside of the University of Washington. In addition, there are currently more than 10 followers as software developers that follow RAT 3.0's continuous development on GitHub. Secondly, RAT 3.0 has experienced independent set up in multiple river basins by researchers at the University of Washington who

were not RAT developer (Sanchit Minocha) – see for example  Darkwah et al. (2023) for Columbia river basin,  Das et al. (2023) for Cumberland river basin and Suresh et al. (2023) for Kerala river basins. Given that it was possible to set up RAT 3.0 in a few weeks compared to a few months that typically took for version 2.0 and 1.0, we are confident that RAT 3.0 is the most user-friendly version of RAT that is available.

Specifically, when we refer to RAT 3.0 as "user-friendly" in the manuscript we intend to convey that individuals with varying levels of expertise, particularly those outside the traditional hydrology community, can easily utilize the tool. The emphasis is on minimal input requirements and the need for only basic knowledge of hydrology and remote sensing. In this context, we highlighted the potential benefits for non-hydrology communities, such as farmers and fishing communities, to access and utilize RAT 3.0. By "community," we are referring to the boards or departments responsible for the well-being of these communities, as they can leverage RAT 3.0 in a user-friendly manner. To make this clear we have reframed the statement in section 2.2 as follows:

"Input requirements have been significantly reduced in RAT 3.0, making it easier for non-hydrology communities such as stakeholders representing farmers and fishing communities to operate the tool."

We have also added the following in Discussion and Conclusion section to make it clear for the readers that why the authors believe RAT 3.0 is user friendly.

"The fact that RAT 3.0 is applied in various river basins, coupled with the reduced setup time to a week from 3-4 months (for RAT 2.0), highlights the scalability and user-friendly architecture of RAT 3.0."

To address reviewer's concern about the distinction between developers and stakeholders, we have revised the manuscript, wherever applicable, to clarify that the term "user-friendly" primarily applies to the ease of executing and utilizing RAT 3.0 for obtaining reservoir operations data. We hope these modifications provide a clearer understanding of the intended audience and the user-friendliness of RAT 3.0.

Computational resources? What kind of computational resources are required to install/implement RAT 3.0? This can be a significant barrier both for computer hardware/infrastructure as well as human capacity. Specifying this in the manuscript would be helpful.

**Response:** Thank you for your valuable feedback regarding the computational resources required for installing and implementing RAT 3.0. To clarify, the primary requirement for setting up and executing RAT 3.0 is a Unix operating system. This restriction is due to the VIC hydrologic model, which is currently available exclusively for Unix operating systems. We have updated the manuscript to include the following information in the Discussion and Conclusion section (other than section 2.2), to addressing the potential barrier related to the choice of operating system. The specific CPU requirements will obviously define the time of computation, but RAT 3.0 is not limited by any specific CPU requirements. Please note that we plan to address

these CPU requirements and data logistics for river basins of varying size in a forthcoming paper that is under preparation for submission to Environmental Modeling and Software. We have addressed this issue of CPU resources in the Discussion and Conclusion section of the manuscript as follows:

"In this paper, we have described the redesigned software architecture of RAT 3.0 that can be executed on any Unix operating system and even on supercomputers by making use of parallelization. The operating system constraint arises from the VIC hydrological model's exclusive compatibility with Unix OS. Consequently, a standard laptop with 8 GB RAM, 4 cores, and a 512 GB hard disk is sufficient for running RAT 3.0, with the computational time contingent on the size of the river basin."

References:

Biswas, N. and F. Hossain (2022) A Multidecadal Analysis of Reservoir Storage Change in Developing Regions, Journal of Hydrometeorology, Vol 23(1), pp. 71-85

Bonnema, M. and Hossain, F. (2017). Inferring reservoir operating patterns across the Mekong Basin using only space observations, *Water Resources Research*, *53*, 3791–3810, (http://dx.doi.org/10.1002/2016WR019978).

Darkwah, G, Hossain F., Tchervenski V., Holtgrieve G., Graves D., Seaton C., Minocha S., Das P., Khan S., Suresh S.(2023) Reconstruction of the Hydro-Thermal History of Regulated River Networks Using Satellite Remote Sensing and Data-driven Techniques, Earth's Future (In review).

Das, P., Hossain, F., Khan, S., Biswas, N. K., Lee, H., Piman, T., Meechaiya, C., Ghimire, U. and Hosen, K. (2022). Reservoir Assessment Tool 2.0: Stakeholder driven improvements to satellite remote sensing based reservoir monitoring. *Environmental Modelling & Software*, *157*, 105533. https://doi.org/10.1016/j.envsoft.2022.105533

Das, P., Hossain F., Minocha S., Suresh S., Darkwah G., Andreadis K., Lee H., Laverde M., Oddo P.(2023) ResORR: A Globally Scalable and Satellite Data-driven Algorithm for River Flow Regulation due to Reservoir Operations, Environmental Modelling and Software (In review)

Schaperow, J. R., Li, D., Margulis, S. A., and Lettenmaier, D. P. (2021). A near-global, high-resolution land surface parameter dataset for the variable infiltration capacity model. *Scientific Data*, *8*(1). https://doi.org/10.1038/s41597-021-00999-4

Suresh, S., Hossain F., Minocha S., Das P., Khan S., Lee H., Andreadis K. and Oddo P.(2023). Satellite-based Tracking of Reservoir Operations for Flood Management during the 2018 Extreme Weather Event in Kerala, India, Remote Sensing of the Environment (In review).

---

## Author Response (AR2)

**RESPONSE TO THE EDITOR**

On 18 January, 2024, Topic editor Dr. Lee provided us with the following directions for revision:

*"The authors seem making their best efforts to improve the quality of the manuscript following the comments from two reviewers. The improvement must be judged by two reviewers again to justify the responses from the authors. Therefore, this TE suggests reconsidering after major revisions."*

**Our response:** We appreciate the TE's positive assessment. We wish to point out that we have received a re-review of the earlier revised manuscript only from ONE reviewer (Reviewer#1). In the above, the TE mentions 'two' reviewers. We have searched hard and we could not find the re-review by Reviewer#2 Dr. Jim Nelson. We would also like to note that Reviewer#1 (Report#1) mentions in their decision ***"For final publication, the manuscript should be accepted as is".***

Reviewer#1 has provided only one comment according to which we have formulated our revision and provided our response to that comment (see below).

The key additional work we have undertaken to address the reviewer#1's concerns are as follows:

- ❖ We have improved the quality of figures.
- ❖ We have elaborated the captions for the Tables.
- ❖ We have made formatting changes to improve presentation.

In the section below, our response to the reviewer#1 comment is shown in blue while the reviewer's comments are in black.

**SPECIFIC RESPONSES TO REVIEWER#1 (Report#1)**

**Comment One:** *"For final publication, the **manuscript should be accepted as is.**"*

**Our response:** Thank you for your endorsement of our paper and our revision efforts earlier.

**Comment Two:** *"The paper effectively highlights the key technological advancements in RAT 3.0 that make it more accessible to wider community of users and researchers for reservoir studies. I appreciate the authors' prompt response and **am pleased to accept the manuscript**. A gentle reminder: please address the formatting issues and specific details (e.g., line 408, Table caption, Figure quality), as they impact the overall quality of the paper."*

**Our Response:** Thank you for the comment and bringing our attention to the formatting issues in the manuscript. We have made formatting changes wherever required to maintain the uniform and consistent formatting throughout the manuscript. Also, figures with less than 300 dpi have been replaced with higher quality figures (figure number 11 and 14) as follows:

[Figure]

Figure 11

[Figure]

Figure 14

We have also edited the Table captions (Table 1,2 and 3) to elaborate the information about the data it contains.

[revised manuscript text omitted]

We have provided a Track change manuscript for the reviewer and TE to verify that we have made all those changes to improve the quality of the manuscript.

**SPECIFIC RESPONSES TO REVIEWER#2 (Dr. Jim Nelson)**

Reviewer#2 had no review of the revised manuscript. His earlier decision based on his review submitted on Nov 11 2023 was ***"accepted subject to minor revisions."***

Our response: NONE (because there is no Report#2 to go by). However we sincerely appreciate Dr. Nelson's clear endorsement of our work to develop a globally scalable reservoir tracking tool that lowers the barrier of entry for the community to replicate and reproduce independently for their own geo-scientific model development.